# Experimental Design for Linear Functionals in Reproducing Kernel Hilbert Spaces

**Mojmír Mutný**
ETH Zürich
mojmir.mutny@inf.ethz.ch

**Andreas Krause**
ETH Zürich
krausea.ethz.ch

## Abstract

Optimal experimental design seeks to determine the most informative allocation of experiments to infer an unknown statistical quantity. In this work, we investigate the optimal design of experiments for *estimation of linear functionals in reproducing kernel Hilbert spaces (RKHSs)*. This problem has been extensively studied in the linear regression setting under an estimability condition, which allows estimating parameters without bias. We generalize this framework to RKHSs, and allow for the linear functional to be only approximately inferred, i.e., with a fixed bias. This scenario captures many important modern applications, such as estimation of gradient maps, integrals, and solutions to differential equations. We provide algorithms for constructing bias-aware designs for linear functionals. We derive non-asymptotic confidence sets for fixed and adaptive designs under sub-Gaussian noise, enabling us to certify estimation with bounded error with high probability.

## 1 Introduction

Optimal Experimental Design (OED) aims to determine data collection schemes – *designs* – to efficiently estimate unknown quantities of interest given limited resources (Chaloner and Verdinelli, 1995). As common, we model experiments via an *oracle* that yields a (noisy) response to a given input. A design is usually either an (adaptive) policy for querying the oracle, or a (nonadaptive) fixed allocation of query budget to different oracle inputs. OED has a rich history and close relations to the field of bandits (Szepesvari and Lattimore, 2020) and active learning (Settles, 2009).

We consider the regression setting, where observations at a fixed input $x \in \mathcal{X}$ can be obtained via the noisy oracle by:

$$y = \theta^\top \Phi(x) + \epsilon \text{ where } x \in \mathcal{X} \tag{1}$$

and $\cdot^\top \cdot$ denotes the inner product in a Hilbert space $\mathcal{H}_\kappa$, $\epsilon$ is independent, sub-Gaussian noise with *known* variance proxy $\sigma^2$, and $\theta$ is a bounded element from a separable reproducing kernel Hilbert space $\mathcal{H}_\kappa$ from kernel $\kappa$, with a bound $\theta^\top \mathcal{V}_0 \theta \le \lambda^{-1}$, where $\mathcal{V}_0 : \mathcal{H}_k \to \mathcal{H}_k$ is a positive definite operator. [1] Depending on whether $\lambda$ is known or unknown, we will propose different estimators. The norm constraint (under $\mathcal{V}_0$) can be interpreted as a bound on the total effect and its parameters if the Hilbert space is finite dimensional. For infinite dimensional spaces this assumption can model the additional constraint on regularity. For example, for kernels that induce Sobolev spaces, it explicitly bounds the total squared derivatives of functions – quantities known to govern complexity of approximation difficulty. The exact bound depends on the specific definition of $\kappa$, but if two spaces are isometrically isomorphic, the difference can be absorbed to $\mathcal{V}_0$ to induce the desired prior regularity, e.g., the regularity of the derivatives. Here we assume that kernel $\kappa$ and the operator $\mathcal{V}_0$ are known due to prior analysis and/or first principles modeling of the system.

In contrast to the classical ED task of estimating $\theta$ (see Fedorov and Hackl, 1997), we are interested in estimating a *projection* of $\theta$. Namely, let $\mathbf{C} : \mathcal{H}_\kappa \to \mathbb{R}^p$ be a *known linear operator*. The map $\mathbf{C}$ is

---

[1]In most cases, but not all, $\mathcal{V}_0$ is chosen to be the identity.

36th Conference on Neural Information Processing Systems (NeurIPS 2022).

such that $\mathbf{C}^\top\mathbf{C}$ is full rank $p$, where $\mathbf{C}^\top$ denotes the adjoint. Our goal is to identify an estimate of $\mathbf{C}\theta$ efficiently, i.e., with low query complexity or with a maximal reduction of uncertainty given a fixed query budget $T$.

The formalism above captures, or occurs as a subroutine in, numerous practical problems. For example, *evaluation* at specific target points, *integration* and *differentiation* are all linear operators, among many other useful linear functionals. Other examples include ordinary or partial differential equations operators, spectral transforms, and stability metrics from control engineering. In Section 7 we detail several example applications.

A naive approach would be to first obtain an estimate $\hat\theta$ of $\theta$, and compute $\mathbf{C}\hat\theta$. However, the appeal of estimating linear functionals *directly* is that the number of unknowns of interest may be *much lower* than for the original overall unknown element $\theta$ (which might even be infinite-dimensional). Consequently, we would hope that the query complexity of reducing the variance of the estimate $\mathbf{C}\theta$ scales in the dimension of the range of $\mathbf{C}$, which is $p$. For example, when focusing on finite-dimensional RKHSs, the operator $\mathbf{C} : \mathbb{R}^m \to \mathbb{R}^p$, where $m$ is the dimension of $\mathcal{H}_\kappa$, becomes a matrix. In this work, we study the cases where $p < m$, and show that the estimation error can indeed scale with $p$, and the geometry of the query set $\{\Phi(x)|x \in \mathcal{X}\}$. The estimation *bias* plays a central role in this work as for very large $m$ estimating $\mathbf{C}\theta$ might not be possible up to any precision. The difficulty depends among other things the richness of the query set $\mathcal{X}$, as we will detail later in Sec. 3.

**Sequential Experiment Design** Apart from classical experiment design, where we first commit to what set of queries $x$ we choose, referred to as a *fixed design*, we also consider sequential design, where the selected queries depend on past observations. Suppose for example, we are gathering data to test whether the null $H_0$: $\theta^\top\Phi(x) \geq 0$ for all $x \in \mathcal{X}$ or otherwise. We can incrementally gather evidence and check whether the null hypothesis has already been rejected. As our data depends on prior evaluation points it forms an *adaptive design*. In this work, we develop confidence sets for both fixed and *adaptive designs* – which are of paramount importance in the context of sequential experiment designs, e.g., to define stopping rules of adaptive hypothesis testing problems.

**Contributions** **A)** We consider objectives for experiment design for linear functionals in general RKHS spaces, which carefully take the bias of the estimator into account. **B)** We provide bounds on the query complexity required to reach $\epsilon$ accuracy to estimate linear functionals with high probability. **C)** We construct novel non-asymptotic confidence sets for linear estimators of linear functionals of RKHS elements, both for *fixed* and *adaptive* designs, where queries are *independent* of previous noise realizations, and where they are not, respectively. **D)** We demonstrate the improved inference error due to specially defined designs and new confidence sets on the problems of learning differential equations, linear bandits, gradient maps estimation, and stability verification of non-linear systems.

## 2  Background and Related Work

**Linear Estimators** Let $\mathcal{S} \subset \mathcal{X}$ be a finite set of selected evaluations s.t. $|\mathcal{S}| = n$, potentially repeated. We focus on linear estimators of the form $\mathbf{L} : \mathbb{R}^n \to \mathbb{R}^p$, where $\mathbf{L}y$ is estimating $\mathbf{C}\theta$. An *estimator* is understood here as the algorithm to find the random *estimate* $\mathbf{L}y$. Notice that given $\mathcal{S}$, $\mathbf{L}$ is not a random quantity; the randomness rather comes from the realizations $y$. To choose the estimator $\mathbf{L}$, one classically looks at the second moment of the residuals $\mathbf{C}\theta - \mathbf{L}y$, $\mathbf{E}(\mathbf{L}) = \mathbb{E}[(\mathbf{C}\theta - \mathbf{L}y)(\mathbf{C}\theta - \mathbf{L}y)^\top]$, where the argument signifies how the random variable $y$ is transformed (noise is averaged)

$$\mathbf{E}(\mathbf{L}) = \sigma^2\mathbf{L}\mathbf{L}^\top + (\mathbf{L}\mathbf{X} - \mathbf{C})\theta\theta^\top(\mathbf{L}\mathbf{X} - \mathbf{C})^\top, \quad \text{where} \quad \mathbf{E}(\mathbf{L}) \in \mathbb{R}^{p\times p}, \tag{2}$$

and the matrix $\mathbf{X}$ contains stacked evaluation functionals of the RKHS $\mathbf{X}_{i,x} = \Phi_i(x)$ for $x \in \mathcal{S} \subseteq \mathcal{X}$. We will then seek a way to transform $y$ via estimator $\mathbf{L}$ such that the second moment of residuals is minimized in certain sense.

**Importance of Bias and RKHS** Classical experimental design studies estimation of $\mathbf{C} : \mathbb{R}^m \to \mathbb{R}^p$ where the RKHS is *finite* dimensional (Pukelsheim, 2006) and $\mathbf{X} \in \mathbb{R}^{n\times m}$. On top of that, they consider only estimators which are *unbiased* for *any* $\theta$, in other words, $\mathbf{L}\mathbf{X} = \mathbf{C}$. While, with finite dimensions, this simplification might be reasonable, for infinite dimensional RKHSs, bias is *inevitable* and must be controlled. Consider the case of *estimating the gradient* of a continuous function $f$. We can nearly never learn it up to arbitrary precision from noisy point queries. However, estimation up to a small error is always possible and sufficient. Estimating $\nabla_x f(x)$ from points close to $f(x)$ will incur small bias but larger variance as the change in $f$ compared noise $\epsilon$ is small. Hence balancing these two sources of error is crucial for an informative design in RKHSs.

**Experiment Design: Classical Perspective**  Consider an $m$ dimensional version of the model in (1). Then, among all unbiased estimators, linear least-squares estimators minimize (2) under the Löwner order due to the famed Gauss-Markov theorem. Unbiasedness in this form is synonymous with *estimability*, which means that by repeating the evaluation in $\mathbf{X}$, arbitrary precision can be reached (Pukelsheim, 2006). The second moment of residuals then becomes $\mathbf{W}_\dagger^{-1} = \mathbf{C}\mathbf{V}^\dagger\mathbf{C}^\top$, where $\cdot^\dagger$ denotes a generalized pseudo-inverse and $\mathbf{V} = \mathbf{X}^\top\mathbf{X}$. The matrix $\mathbf{W}$ is often referred to as the *information matrix*. Gaffke (1987) and Krafft (1983) note that estimability implies that $\mathbf{W}_\dagger$ is non-singular. We relax this condition and allow the estimator to have a *bias*; this means that we cannot in general reduce the error arbitrarily by repeating the measurements. In fact, our extension uses the fact that $\mathbf{W}_\dagger$ as defined above will not be singular even if estimability condition is not satisfied. We will show that the matrix $\mathbf{W}_\dagger$ can still play the role of the information matrix.

**Experiment Design: Modern Challenges**  Mutný et al. (2020) use experimental design to estimate the Hessian of an unknown RKHS function, while Kirschner et al. (2019) use it to estimate the gradient of it for use in Bayesian optimization. Perhaps most related, Shoham and Avron (2020) study over-parametrized experimental design for one-shot active deep learning and analyze the bias in connection to experiment design, similarly as in the seminal work of Bardow (2008). They do not consider bias arising from the limited design space nor do they treat linear functionals. More broadly, the uncertainty propagation is studied in the Bayesian framework in the field of *probabilistic numerics* for linear and nonlinear operators (see Cockayne et al., 2017; Owhadi and Scovel, 2016, and citations therein).

**Confidence sets**  Unlike classical statistics, our focus is on non-asymptotic confidence sets on regression estimators. They can be found in, e.g., Draper and Smith (2014) and Abbasi-Yadkori et al. (2011), for fixed and adaptive designs, respectively. Our goal is to define the confidence sets in the appropriate norm such that their width scales with the dimension of the range of $\mathbf{C}$, i.e., $p$, and the geometry of $\{\Phi(x)|x \in \mathcal{X}\}$, but not directly $\dim(\mathcal{H}_\kappa)$ nor number of data $T$. Mutný et al. (2020) derive *non-asymptotic* confidence sets that scale in $p$, but grow with the number of points $T$ as they do not use the appropriate norm (see Appendix B.4). Similarly, Khamaru et al. (2021) study estimators for adaptively collected data, and propose *asymptotic confidence* sets for their estimators. Without a specific condition, their confidence sets can grow with $T$, however, they consider more general noise distributions apart from sub-Gaussian as we do here.

## 3 Estimation and Bias

In this section, we motivate the linear estimators and identify *information matrices*. Information matrices are the inverses of second moment matrices of residuals $\mathbf{E}$ as in Eq. (2). They are an important object in the analysis of the error of the estimators and their confidence sets. Also, they depend on the evaluations that define the estimator $\mathbf{L}$, $\mathbf{X}$. *Maximizing the information matrices* as a function of the chosen observations and their proportions $\mathbf{X}$ gives rise to *optimal experimental designs*.

Further, we identify quantities that influence the bias of the estimators. We study two estimators: the least-norm estimator (interpolation), when the bound on $\|\theta\|_{\mathcal{V}_0}$ is unknown but finite, or the ridge regularized least squares estimator, where $\|\theta\|_{\mathcal{V}_0} \leq \lambda^{-1}$ is known. Both estimators are motivated as minimizing the error residuals $\mathbf{E}$ in trace norm under these two bound assumptions.

### 3.1 Estimators

**Interpolation**  As apparent from Eq. (2), without the knowledge of an explicit bound on the norm $\theta$, the worst case over $\theta$ causes the optimal estimator $\mathbf{L}$ to minimize *only* the trace of the bias. This leads to minimization of $(\mathbf{C} - \mathbf{L}\mathbf{X})\mathcal{V}_0^{-1/2}$ in Frobenius norm, leading to the familiar *interpolation estimator*,

$$\mathbf{C}\hat{\theta} := \mathbf{L}_\dagger y = \mathbf{C}\mathcal{V}_0^{-1}\mathbf{X}^\top\mathbf{K}^{-1}y, \quad \text{where} \quad \mathbf{K} = \mathbf{X}\mathcal{V}_0^{-1}\mathbf{X}^\top \tag{3}$$

The second moment of residuals can then be expressed as

$$\mathbf{E}(\mathbf{L}_\dagger) \preceq \underbrace{\sigma^2\mathbf{C}\mathcal{V}_0^{-1}\mathbf{X}^\top\mathbf{K}^{-2}\mathbf{X}\mathcal{V}_0^{-1}\mathbf{C}^\top}_{\text{variance}} + \underbrace{\frac{1}{\lambda}\mathbf{C}\mathcal{P}_\mathbf{X}\mathbf{C}^\top}_{\text{bias}},$$

where $\mathcal{P}_\mathbf{X} = \mathcal{V}_0^{-1/2}(\mathcal{V}_0^{-1/2}\mathbf{X}^\top\mathbf{K}^{-1}\mathbf{X}\mathcal{V}_0^{-1/2} - \mathcal{I})\mathcal{V}_0^{-1/2}$ is a scaled projection matrix. Unlike as in the classical ED treatment, the second moment has two terms: bias and variance. If the span of the scaled projection operator lies in the null space of $\mathbf{C}$, the bias (classically) vanishes. To control the error of estimation, we need to control both terms. For the special case of interpolation estimator,

we will control the second term separately using a *bias condition*, and the variance term will be controlled by the *information matrix* $\mathbf{W}_\dagger$ as we will see in Section 4

$$\mathbf{W}_\dagger(\mathbf{X}) = (\mathbf{C}\mathcal{V}_0^{-1}\mathbf{X}^\top\mathbf{K}^{-2}\mathbf{X}\mathcal{V}_0^{-1}\mathbf{C}^\top)^{-1}. \tag{4}$$

**Regularized Regression**    The ridge regularized estimator is motivated by using $\theta^\top\mathcal{V}_0\theta \leq \lambda^{-1}$, and minimizing then the trace of this upper bound, leading to an estimator,

$$\mathbf{C}\hat{\theta}_\lambda := \mathbf{L}_\lambda y = \mathbf{C}\mathcal{V}_0^{-1}\mathbf{X}^\top(\lambda\sigma^2\mathbf{I} + \mathbf{K})^{-1}y. \tag{5}$$

Like above, we can give an upper bound on the second moment of the residuals. Conveniently, this estimator automatically balances the error due to variance and bias in one term: $\mathbf{E}(\mathbf{L}_\lambda) \preceq \mathbf{C}\theta\theta^\top\mathbf{C}^\top - \mathbf{C}\mathcal{V}_0^{-1}\mathbf{X}^\top(\mathbf{I}\sigma^2\lambda + \mathbf{K})^{-1}\mathbf{X}\mathcal{V}_0^{-1}\mathbf{C}^\top$. Using the matrix inversion lemma, we can express the above bound in a more concise form, and subsequently its inverse motivates the definition of the information matrix for the regularized estimator,

$$\mathbf{W}_\lambda(\mathbf{X}) = \sigma^{-2}(\mathbf{C}(\sigma^2\lambda\mathcal{V}_0 + \mathbf{X}^\top\mathbf{X})^{-1}\mathbf{C}^\top)^{-1}. \tag{6}$$

### 3.2   Design objectives: Scalarization

The information matrices such as in (4) and (6) represent the inverse of estimation error, and the goal of optimal design is to maximize them (hence minimizing the error) with a proper choice of $\mathbf{X}$. As the Löwner order is not a total order, we need to resort to some scalarization of the information matrices, thus we solve $\max_\mathbf{X} f(\mathbf{W}(\mathbf{X}))$, where $f$ is the scalarization $\mathbb{R}^p \to \mathbb{R}$. We focus on two common forms of scalarization and refer to them as $E$- and $A$-design (Pukelsheim, 2006).

- $f_E(\mathbf{W}) = \lambda_{\min}(\mathbf{W})$ – when constructing non-asymptotic high probability confidence sets;
- $f_A(\mathbf{W}) = 1/\operatorname{Tr}(\mathbf{W}^{-1})$ – when minimizing mean squared error of estimation.

Other popular criteria include $D$, $V$, and $G$-designs (Chaloner and Verdinelli, 1995), which we do not consider due to space considerations, but can be equivalently used should the experimenter have a reason for it. If $p = 1$, $\mathbf{C}$ is an element in $\mathcal{H}_k$ and the design problem degenerates into a special case known as c-optimality due to Elfving (1952) and, as $\mathbf{W} \in \mathbb{R}$, no scalarizations are needed.

**Robust Designs**    The linear functionals are sometimes unknown, or parametrized by an *unknown* parameter $\gamma$ as $\mathbf{C}_\gamma$, where $\gamma$ belongs to a known set $\Gamma$. If we were to construct a design that has low estimation error for the worst case selection of $\gamma$, we can maximize the *information* in the following worst case metric $\max_\mathbf{X} \inf_{\gamma\in\Gamma} f\left(\left(\mathbf{C}_\gamma\mathcal{V}_0^{-1}\mathbf{X}^\top\mathbf{K}^{-2}\mathbf{X}\mathcal{V}_0^{-1}\mathbf{C}_\gamma^\top\right)^{-1}\right)$, for the interpolation estimator (and analogously for ridge regression). If the original function $f(\mathbf{W})$ is concave, then so is the function defined as the infimum over the compact index set $\Gamma$, which is true for the design criteria considered in this work (Boyd and Vandenberghe, 2004).

## 4   Fixed Designs and their Confidence Sets

In RKHS spaces, especially infinite dimensional ones, the *estimability* without bias is too restrictive. Given a finite evaluation budget $T$, we can only construct a discrete design $\mathbf{X} : \mathcal{H}_\kappa \to \mathbb{R}^T$, and there are many practical examples, where, given a finite query budget, $\mathbf{C}\theta$ cannot be learned to arbitrary precision, most prominently gradients and integrals among many others.

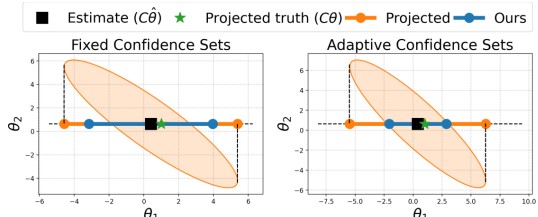

**Estimation with Bias and Interpolation Estimator**    Our goal is to establish a condition on the design space $\mathbf{X}$ such that the estimation with the interpolation estimator is possible up to a certain bias $\epsilon$ measured under the Euclidean norm. We measure the bias scaled by the magnitude of the Frobenius norm of the estimator and call this the relative $\nu$-bias. This condition will allow us to balance the error due to the bias and noise.

Figure 1: A two dim. example. Left: fixed design with our and projected confidence sets from two dimensions. Right: Adaptive confidence sets compared with projected ones due to Abbasi-Yadkori and Szepesvari (2012). In this example $\mathbf{C} = (1, 0)$ and $\theta = 0$.

**Definition 1** (Relative $\nu$-bias). Let $\mathbf{C} : \mathcal{H}_\kappa \to \mathbb{R}^k$. The estimator $\mathbf{L}_\dagger$ on the design space $\mathbf{X}$ is said to have *relative $\nu$-bias* if

$$\left\|(\mathbf{C} - \mathbf{L}_\dagger\mathbf{X})\mathcal{V}_0^{-1/2}\right\|_F^2 \leq \nu^2 \left\|\mathbf{L}_\dagger\right\|_F^2. \tag{7}$$

The $\|\cdot\|_F$ corresponds to the Frobenius norm of the maps $\mathcal{H}_\kappa \to \mathcal{H}_\kappa$. Due to the cyclic property of the trace, we can take the adjoint of the operator and calculate the quantity by taking a trace of $p \times p$ matrix instead. If $\nu = 0$ and $\dim(\mathcal{H}_\kappa) = m$, as we show in Lemma 1 in the Appendix, this is equivalent to the classical *estimability* condition due to Pukelsheim (2006). The left hand side corresponds to the *classical bias* $\mathrm{bias}(\mathbf{L}_\dagger y) = \|\mathbb{E}[\mathbf{L}_\dagger y] - \mathbf{C}\theta\|_2$ of an estimator $\mathbf{L}_\dagger$ under the Frobenius norm. This is exactly the quantity that $\mathbf{L}_\dagger$ minimizes. The *classical bias* cannot be improved by repeated measurements or allocations thereof. Using a relation which we show formally in Proposition 5 in Appendix B, we can bound the $\mathrm{bias}(\mathbf{L}_\dagger y) = \|\mathbb{E}[\mathbf{L}_\dagger y - \mathbf{C}\theta]\|_2 \leq \lambda_{\min}(\mathbf{W}_\dagger)^{-1/2}\nu/\sqrt{\lambda}$.

To check whether the condition (7) is satisfied given a design space $\mathbf{X}$, one needs to evaluate the trace. Finding the value of $p \times p$ matrix, however, depends strongly on the form of the operator $\mathbf{C}$, and a general recipe cannot be provided. Due to Riesz's representer theorem, and fact that the range of $\mathbf{C}$ is finite-dimensional, this is always possible. For example, for an integral operator $\int \Phi(x)^\top \cdot q(x)dx$, using shorthand $v = \mathbf{K}^{-1}y$, it holds that $\mathbf{L}_\dagger y = \sum_{i=1}^n \int \kappa(x, x_i)v_i dx$ where we used $n$ points. Notice that we used only evaluations of kernel $\kappa$ to do this calculation. Also notice that calculation of (7) reduces to the calculation of the maximum mean discrepancy (Gretton et al., 2005) between the empirical distribution (defined on $\mathbf{X}$), and $q(x)$ for this example. The general recipe is to (locally) project the Hilbert space on a truncated finite-dimensional basis $\Phi_m(x)$ with size $m$ and, in that case, the check involves a solution to a linear system, but this is not always necessary.

## 4.1 Confidence sets

To construct confidence sets for both estimators considered, we split them into two categories: *fixed designs*, where the evaluation queries do not depend on the observed values $y$, and *adaptive designs*, where the queries $\Phi(x_i)$ *may depend* on prior evaluations $y_{i-1}, \ldots y_1$. Proofs are in Appendix B.

**Theorem 1** (Interpolation – Fixed Design). *Under the regression model in Eq. (1) with $T$ data point evaluations, let $\hat{\theta}$ be the estimate as in (3). Let $\mathbf{X}$ satisfy the bias from Def. 1 with $\mathbf{W}_\dagger = (\mathbf{C}\mathcal{V}_0^{-1/2}\mathbf{V}^\dagger \mathcal{V}_0^{-1/2}\mathbf{C}^\top)^{-1}$. Then,*

$$\mathrm{P}\left(\left\|\mathbf{C}(\hat{\theta} - \theta)\right\|_{\mathbf{W}_\dagger} \geq \sigma\sqrt{\xi(\delta)} + \frac{\nu}{\sqrt{\lambda}}\right) \leq \delta, \tag{8}$$

*where $\mathbf{V}^\dagger = \mathcal{V}_0^{-1/2}\mathbf{X}^\top(\mathbf{X}\mathcal{V}_0^{-1}\mathbf{X}^\top)^{-2}\mathbf{X}\mathcal{V}_0^{-1/2}$, and $\xi(\delta) = p + 2\left(\sqrt{p\log(\frac{1}{\delta})} + \log\left(\frac{1}{\delta}\right)\right)$.*

Notice that in order to balance the source of the error due to bias and variance with high probability, we need to match $\sigma\sqrt{\xi(\delta)} \approx \frac{\nu}{\sqrt{\lambda}}$. Repeating the queries in $\mathbf{X}$ $T$ times reduces $\sigma$ by $1/\sqrt{T}$ but leaves the bias $\nu$, as well as $\mathbf{W}_\dagger$, unchanged. This is because with the interpolation estimator the noisy repeated queries are averaged, which can be interpreted as a reduction in variance. Hence, by balancing $\nu/\sqrt{\lambda}$ with $\sigma\sqrt{\xi(\delta)}/\sqrt{T}$, we balance the bias and variance such that they are of the same magnitude. It does not make sense to repeat measurements more times if the bias dominates the error of estimation. A detailed example of estimating the gradient with fixed bias is given in Sec. 7.2.

We derive confidence sets for the regularized estimator of Eq. (5), albeit without the relative bias.

**Proposition 2** (Regularized estimate – Fixed Design). *Under the model in Eq. (1) with $T$ data point evaluations, let $\hat{\theta}_\lambda$ be the regularized estimate as in (5). Then* $\mathrm{P}\left(\left\|\mathbf{C}(\hat{\theta}_\lambda - \theta)\right\|_{\mathbf{W}_\lambda} \geq \sqrt{\xi(\delta)} + 1\right) \leq \delta$, *where $\xi(\delta) = p + 2\left(\sqrt{p\log(\frac{1}{\delta})} + \log\left(\frac{1}{\delta}\right)\right)$.*

Notice that, since the regularized estimator is designed to balance the bias and variance automatically, we do not need to specifically control the bias, which is contained within the information matrix $\mathbf{W}_\lambda$. Despite this elegant property, the regularized estimator involves a more challenging analysis. The main motivation to study the interpolation estimator is to understand the $l_2$ error as we show in Section 6.

## 5 Adaptive Design and Confidence Sets

To provide confidence sets for adaptively collected data, we need to project the data in $\mathbf{X}$ onto $\mathbf{C}$, where we will denote the projection by $\mathbf{Z}$ further on. With the data projected, we can reason about the reduction of the uncertainty of $\mathbf{C}\theta$ for each point separately, since to each $\Phi(x_i)$ we can associate a unique $z_i$ in $\mathbb{R}^p$.

**Definition 2** (Projected data). Let $z(x) \in \mathbb{R}^p$, *projected data*, be a vector field s.t. $\Phi(x)\mathcal{V}_0^{-1/2} = z(x)\mathbf{C}\mathcal{V}_0^{-1/2} + j(x)$, where $x \in \mathcal{S} \subseteq \mathcal{X}$ and $j(x) \in \mathcal{H}_\kappa$, $|\mathcal{S}| = n$, such that $\mathbf{C}\mathcal{V}_0^{-1/2}j(x) = 0$.

Classically, the adaptive confidence sets are understood only for $\mathbf{C} = \mathbf{I}$, i.e., the identity (Abbasi-Yadkori et al., 2011). In fact, we can always derive confidence sets for $\mathbf{C}\theta$ from confidence sets for $\theta$, as they only project the ellipsoid to a smaller dimensional space. However, their resulting size may be unnecessarily large, as the confidence parameter scales as $\mathcal{O}(\dim(\mathcal{H}_k))$ in general (see Figure 1 for a visual example).

The martingale analysis of Abbasi-Yadkori et al. (2011) and de la Peña et al. (2009) specifically assumes that information matrix $\mathbf{V}_t$, (where $\mathbf{C} = \mathbf{I}$) can be additively decomposed to information matrices due to a single evaluation $\mathbf{V}_t = \sum_{i=1}^t \Phi(x_i)\Phi(x_i)^\top$ each at a different time. With the matrix $\mathbf{W}_{\lambda,t}$, this additive decomposition is not always possible. Therefore, to utilize the martingale analysis, which requires this additive property, we consider a different information matrix, which upper bounds $\mathbf{W}_\lambda$. The information matrix we use, $\mathbf{\Omega}_\lambda$, is constructed from the projections $z(x_i)$, which have the necessary additive property. It gives rise to confidence sets, where, under the ellipsoidal norm, their size scales as $\Theta(p)$. The estimation error depends on $\mathbf{\Omega}_\lambda$ still, but under this norm, the confidence parameter and information matrix are decoupled in a similar way as for the fixed design.

**Theorem 3** (Ridge estimate – Adaptive Design). *Under the regression model in Eq.* (1) *with $t$ adaptively collected data points, let $\hat{\theta}_{\lambda,t}$ be the regularized estimate as in* (5). *Further, assume that $\mathbf{Z}$ is as in Def.* 2 *where $\mathbf{X}_t = \mathbf{Z}_t\mathbf{C} + \mathbf{J}_t\mathcal{V}_0^{1/2}$. Then for all $t \geq 0$,*

$$\left\|\mathbf{C}(\hat{\theta}_t - \theta)\right\|_{\mathbf{\Omega}_{\lambda,t}} \leq \sqrt{2\log\left(\frac{1}{\delta}\frac{\det(\mathbf{\Omega}_{\lambda,t})^{1/2}}{\det(\lambda\mathbf{S})^{1/2}}\right) + 1} \tag{9}$$

*with probability $1 - \delta$, where $\mathbf{\Omega}_{\lambda,t} = \frac{1}{\sigma^2}\mathbf{Z}_t^\top\mathbf{Z}_t + \lambda\mathbf{S}$ and $\mathbf{S} = (\mathbf{C}\mathcal{V}_0^{-1}\mathbf{C}^\top)^{-1}$.*

The matrix $\mathbf{Z}_t$ can be calculated by solving a least-squares problem (projection), whereupon $\mathbf{C}\mathcal{V}_0^{-1/2}\mathbf{J}_t = 0$ as needed by Definition 2. Notice that, on one hand, the above confidence parameter grows only when $\mathbf{Z}$ is large, but at the same time, the ellipsoid shrinks only in that case as well. Also, the ellipsoid above is necessarily smaller than the one with the information matrix $\mathbf{W}_\lambda$ as $\mathbf{W}_\lambda \preceq \mathbf{\Omega}_\lambda$ (Lemma 3 in Appendix). This means we can use the same confidence parameter to give a bound for $\|\cdot\|_{\mathbf{W}_{\lambda,t}}$. Notice that the estimator is the same as before, i.e., $\hat{\theta}_\lambda$, using $\mathbf{X}$ to define the regression, only the information matrix changes. We also present a visual comparison in Figure 1 on a simple example, showing that our fixed and adaptive sets are tighter than projected non-asymptotic sets. Using the shorthand $\beta_t(\delta)$ to define the confidence parameter in (9), we can in fact bound the error as $\|\mathbf{C}(\theta_{\lambda,t} - \theta)\|_2 \leq \lambda_{\min}(\mathbf{\Omega}_{\lambda,t})^{-1/2}\beta_t(\delta)$. It depends on $m = \dim(\mathcal{H}_\kappa)$ only via $\mathbf{\Omega}_{\lambda,t}$. The dependence of the confidence parameter can be at most $\Theta(\sqrt{\log(1+T)p})$, see Lemma 4 in Appendix B. The value of $\lambda_{\min}(\mathbf{\Omega}_{\lambda,t})^{-1}$ depends on the geometry of the set $\{\Phi(x)|x \in \mathcal{X}\}$ and the projection operator $\mathbf{C}$ as we show in Section 6.3. Note that the lower bound of (Szepesvari and Lattimore, 2020, Ex. 20.2.3) does not apply here, since it makes a statement only about the information matrix $\mathbf{W}_\lambda$.

# 6 Convex Relaxations, Geometry and Dimensions

Suppose we are given a candidate set of experiments, i.e., a unique subset of evaluations $\mathcal{S} \subset \mathcal{X}$, $|\mathcal{S}| = n$ and a budget $T$ of total queries. We seek an allocation $\mathbf{X}$, where the rows of $\mathbf{X}$ contain potentially repeated evaluations from $\mathcal{S}$. How many times should we repeat each experiment in order to find $\max_{\mathbf{X}} f(\mathbf{W}(\mathbf{X}))$? To address this, experimental design literature relaxes this discrete optimization problem, and optimizes over fractional allocation $\eta \in \Delta^n$, where the number of repetitions for $\Phi(x_i)$ is recovered by rounding $\lceil\eta_i T\rceil$. With this interpretation, the objectives in Sec. 3.2 can be written as

$$\eta^* = \arg\max_{\eta \in \Delta^n}\left[f(\mathbf{W}_\dagger(\mathbf{D}(\eta)^{1/2}\mathbf{X}_\mathcal{S})) = f(\mathbf{C}(\mathbf{V}_0^{-1}\mathbf{X}_\mathcal{S}^\top\mathbf{D}(\eta)\mathbf{X}_\mathcal{S}\mathbf{V}_0^{-1})^\dagger\mathbf{C}^\top)\right], \tag{10}$$

where $\mathbf{D}(\eta)$ is the diagonalization operator that produces a diagonal matrix with vector $\eta$ on the diagonal and $\mathbf{X}_\mathcal{S}$ contains non-repeated elements in $\mathcal{S}$. We have stated the problem above for the interpolation estimator and $\dim(\mathcal{H}_k) < \infty$, but it naturally generalizes to kernelized estimators, albeit in a less concise form (see Appendix C).

## 6.1 Optimizing allocations: experiment design algorithms

Given a subset $\mathcal{S}$, the problem (10) can be approximately solved using either convex optimization methods or a greedy algorithm. A comprehensive review of methods constructing designs $\eta^*$ and rounding techniques to get $\mathbf{X}$ is *beyond the scope of this work*, and *not the core issue* addressed in this work. We briefly review two versatile approaches for completeness (more details in Appendix C).

**Greedy selection**    Firstly, one can greedily maximize the scalarized information matrix with the update rule $\eta_{t+1} = \frac{t}{t+1}\eta_t + \frac{1}{1+t}\delta_t$, $\delta_t = \arg\max_{x \in \mathcal{X}} f\left(\mathbf{W}_\lambda\left(\frac{t}{t+1}\eta_t + \frac{1}{t+1}\delta_x\right)\right)$, where $f$ refers to the scalarization and $\delta_x$ to the discrete measure corresponding to the feature map $\Phi(x)$. Due to the form of the update rule, $t\eta_t$ is always an integer.

## 6.2 Convex optimization

Alternatively, convex optimization can provably solve the problem to optimality. Specifically, we look for an allocation using convex optimization methods $\max_{\eta \in \Delta^n} f(\mathbf{W}_\circ(\mathbf{D}(\eta)^{1/2}\mathbf{X}))$ where $\circ \in \{\dagger, \lambda\}$. Care needs to be taken when selecting $\mathcal{S}$, as we discuss in Appendix C. The most common algorithms for ODE problems are the Frank-Wolfe algorithm (Todd, 2016) and mirror descent algorithm (Silvey et al., 1978). Additionally, semi-definite reformulation of the above optimization problems can be given as we shown in Appendix C.3. Optimal designs found via convex optimization need to be rounded in practice. State-of-art rounding techniques are discussed by Allen-Zhu et al. (2017) and Camilleri et al. (2021) for finite and infinite dimensional spaces, respectively. The exhaustive cover of rounding techniques is out-of-scope of the current work.

## 6.3 Estimation error and its dimension dependence

If we were to bound the squared error of estimation in high probability, the importance of the scalarization $\lambda_{\min}(\mathbf{W}_\dagger)$ becomes apparent. Using the Cauchy Schwarz inequality, we get

$$\left\|\mathbf{C}(\theta - \hat{\theta})\right\|_2 \leq \lambda_{\min}(\mathbf{W}_\dagger)^{-1/2}\left\|\mathbf{C}(\theta - \hat{\theta})\right\|_{\mathbf{W}_\dagger} \leq \sqrt{\frac{\lambda_{\min}(\mathbf{W}_\dagger(\eta^*))^{-1}}{T}}(\sigma\sqrt{\xi(\delta)} + \nu/\sqrt{\lambda}),$$

where the term due to Proposition 1 scales as $\mathcal{O}(p)$ when properly balanced. Using the optimal allocation $\eta^*$, the number of repetitive evaluations is equal to $\eta^*T$. Inverting the relation above yields a query complexity of the order $T \approx \mathcal{O}(\frac{p}{\epsilon^2}\lambda_{\min}(\mathbf{W}_\dagger(\eta^*)^{-1}))$. The optimal value $\lambda_{\min}(\mathbf{W}_\dagger(\eta^*))$ represents a problem dependent quantity that measures the *difficulty of estimation* and captures the geometry of the set $\{\Phi(x)|x \in \mathcal{S} \subset \mathcal{X}\}$. It cannot be bounded in general, but it has an elegant geometric interpretation. In particular, it corresponds to the square inverse of the *diameter of the largest inscribed ball in the convex hull of symmetrized* $\{\Phi(x)|x \in \mathcal{S}$ *in the range of* $\mathbf{C}$ (Pukelsheim and Studden, 1993).

At first glance, calculating this quantity might seem complicated, but with an example it is apparent. For example, if $\Phi(x) = x$ s.t. $\|x\|_2^2 \leq 1$ (unit $l_2$ ball in $\mathbb{R}^m$) and $\mathbf{C} = v$, where $v$ is a unit vector ($p = 1$), the inverse diameter of largest ball we can inscribe in direction of $v$ inside $\|x\|_2^2 \leq 1$ equates to 1, which is independent of $d$. This is not surprising, since it represents an "easy" design space, where for any direction $v$ one can find an action $x$ aligned with that coincides with the optimal design. This is true even if $\mathbf{C}$ has more rows. On the other hand, if we assume evaluation in the $l_1$ ball $\|x\|_1 \leq 1$ and $\mathbf{C}$ is a vector of ones (again $p = 1$). Then the diameter of the inscribed ball is proportional to the inverse square height of a simplex, namely, $\frac{1}{m}$, despite $p = 1$. Hence, despite the confidence parameter being $\mathcal{O}(1)$, the complexity depends primarily on the geometry of this set.

To give a more exotic example, if $\mathbf{C} = \nabla_x\Phi(x)$, and the design space are points with fixed length steps in all unit derection $\{x|\Phi(x \pm e_i h)\}$, where $h$ is the stepsize and $e_i$ are principal vectors in $\mathbb{R}^d$, we can show that $\lambda_{min}(\mathbf{W}_\dagger)^{-1} \leq dh + \mathcal{O}(h^2)$, leaving the overall complexity to learn a gradient to scale with $d$ instead of the dimensionality of the RKHS which can be infinite. All formal proofs and references are provided in Appendix D.1.

# 7  Applications

We now discuss concrete applications that benefit from our contributions. Details of the experiments, and further applications, e.g., in statistical contamination, can be found in Appendices E and F. We provide information about optimization and rationale in choosing the $\kappa$, and $\mathcal{V}_0$ in C and F respectively.

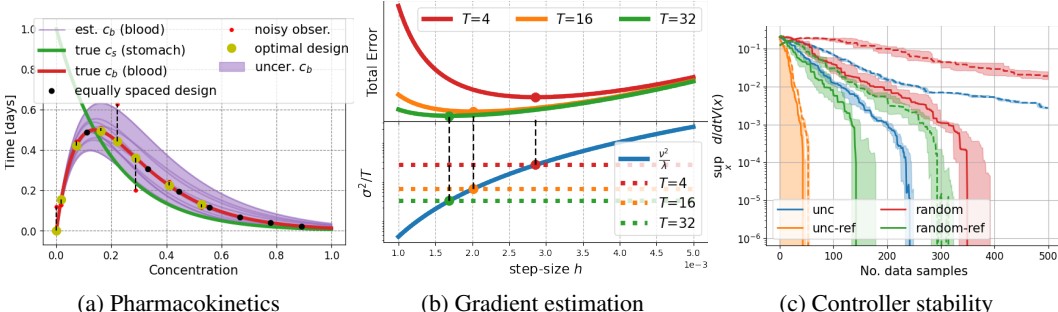

| (a) Pharmacokinetics | (b) Gradient estimation | (c) Controller stability |

Figure 2: Experiments: **a)** *Pharmacokinetics*. We compare the equally spaced design (black) with the optimized design (yellow). The optimized design mimics the classical pharmacokinetics approach of spreading the initial measurements more densely after the initial dose (Gabrielsson and Weiner, 1995). For us, this rule elegantly emerges from first principles. In general, these trajectories follow a decaying pattern as the examples in light purple. The uncertainty in $c_b$ due to the unknown dynamics $\gamma$ is depicted in shaded region. **b)** *Gradient estimation*. The upper plot shows the total error of $\nabla f(x)$ we can certify with high probability as a function of the step-size $h$ for a finite difference design. The minima of these errors exactly correspond to step-sizes derived from the bias-variance trade-off we proposed for the interpolation estimator. With increasing $T$, noise can be reduced more, and hence bias of the design needs to decrease accordingly, hence the decrease in the step size (e.g., red→green). **c)** *Stability:* We report the upper bound on the Lyapunov function in the whole operating domain, as a function of data points. The color coding (explained in the text) represents different data acquisition algorithms while dashed lines correspond to the confidence sets from prior work. Our confidence sets (solid) provide a tighter upper bound as a function of the number of data points, and allows faster termination, which happens when the upper bound is zero.

## 7.1 Linear bandits with finite number of arms

A special but important theoretical consequence of our adaptive confidence set is an improved regret bound for the upper confidence bound (UCB) algorithm (Auer, 2002) for linear bandits when the set of queries $\mathcal{X}$ is finite. The queries are known as actions in the bandit literature (Szepesvari and Lattimore, 2020). The UCB algorithm is a procedure which iteratively queries the action according to $x_t = \arg\max_{x \in \mathcal{X}} \max_{\vartheta \in C_{t-1}} \vartheta^\top \Phi(x)$, where $C_t$ is confidence set for $\theta$ in round $t$. Different to this, in our analysis of the UCB algorithm, we use the confidence sets due to Theorem 3, which are confidence sets for the projections $\Phi(x)^\top \theta$ directly. We construct $|\mathcal{X}|$ confidence sets, one for each of the linear functionals $\mathbf{C}_x = \Phi(x)$, and denote them by $\mathrm{cf}_t(x)$. The UCB algorithm then just becomes $x_t = \arg\max_{x \in \mathcal{X}} \max_{y \in \mathrm{cf}_t(x)} y$. We now analyze its cumulative regret $R_T := \sum_{t=1}^{T}(\Phi(x^*) - \Phi(x_t))^\top \theta$, where $x^* = \arg\max_{x \in \mathcal{X}} \Phi(x)^\top \theta$, and $\theta$ is the true unknown *pay-off vector* of the bandit game.

**Theorem 4.** *Let $\theta \in \mathbb{R}^d$ be an unknown pay-off vector, and a set of actions $x \in \mathcal{X}$ such that $|\mathcal{X}| < \infty$. Then the cumulative regret of the UCB algorithm is bounded by*

$$R_T \leq \mathcal{O}(\sqrt{Td\log(T(|\mathcal{X}| + 1)/\delta)}) \text{ with } 1 - \delta \text{ probability.}$$

Thus, our adaptive confidence sets for linear functionals yield a regret bound that is a factor of $\sqrt{d}$ tighter than the standard bound. Note that this does not break the known lower bound $\mathcal{O}(d\sqrt{T})$ for $\mathcal{X}$ being the unit ball in $\mathbb{R}^d$ from Rusmevichientong and Tsitsiklis (2010), as we gain a logarithmic dependence on $|\mathcal{X}|$. For an $\epsilon$ covering of the extreme points of the unit ball in $\mathbb{R}^d$, one needs $|\mathcal{X}| \propto (1/\epsilon)^d$ points, leading to a regret scaling as $\mathcal{O}(d\sqrt{T}\log(1/\epsilon) + \epsilon T)$. Choosing $\epsilon = 1/T$ would, for a fixed $T$, recover the lower bound up to logarithmic terms. Using standard *doubling-trick* techniques, one can provide anytime result. However, for different query sets (such as the $l_1$ ball), whose extreme points (i.e., number of vertices) are finite (equal to $d$), these results can vastly improve the regret bounds. There are alternative algorithms based on arm-elimination, which achieve this regret bound, e.g., the algorithm of Szepesvari and Lattimore (2020), the SupLinRel algorithm of Auer (2002), and the algorithm of Valko et al. (2014). The proof of Theorem 4 is deferred to Appendix G, but it is a straightforward application of previous results.

The above technique can be extended to infinite dimensional RKHS, where the role of dimension is played by, $\gamma_T$ called maximum information gain (Srinivas et al., 2010), defined in Appendix G. The

remaining challenge is to bound the $\epsilon$-covering of extreme points of the set $\{\Phi(x)|x \in [-1,1]^d\} \subset \mathcal{H}_k$, where $[-1,1]^d$ is the continuous action set. If all evaluation operators have $\|\Phi(x)\|_k = 1$, for all $x \in [-1,1]$, such as for Matérn kernels (Mutný and Krause, 2018), we can use covering numbers of unit balls in these spaces as the upper bound on this number. These spaces are known to be isometrically isomorphic to Sobolev spaces (Wendland, 2004), and the bounds on covering numbers of unit balls are known to be of order $(\frac{1}{\epsilon})^{2d/s}$, where $s \in \mathbb{N}$, $s > d/2$, refers to regularity of the Sobolev space (Cucker and Smale, 2002). This yields a bound on the cumulative regret of the order $\mathcal{O}(\sqrt{C(d)\gamma_T T} \log T)$, where $C(d)$ can depend on dimension.

## 7.2 Gradient maps

Gradients of any order are linear operators. We can express the gradient at $x$ as dimension-wise evaluation of the following operator $\nabla_x(\Phi(x)^\top \theta) = (\nabla_x \Phi(x))^\top \theta =: \mathbf{C}\theta$. Clearly, estimability is nearly always impossible, since any evaluation infinitesimally away from $x$ will be insufficient to eliminate bias. Thus our estimates will invariably be biased for any function with infinite Taylor expansion. Yet, while estimating the gradient from evaluations very close to the original $x$ leads to low bias, it at the same time increases the variance, since the difference in the functional value between the two point evaluations is very small compared to the noise magnitude. Given a finite budget or desired accuracy, the best we can do is to find the best design with optimal bias-variance trade-off given the kernel $\kappa$, budget $T$, and noise variance $\sigma$. We consider a class of parametrized finite difference designs $\{\Phi(x \pm he_i)\}$, where $h$ is the stepsize and $e_i$ are unit vectors. In Fig. 2b, we plot the total error with high probability with the budget $T$ as a function of the step-size $h$. Notice that the lowest error occurs exactly when the variance of observations scaled with the confidence parameter is equal to the relative $\nu$ (two lines cross).

## 7.3 Learning linear ODE solutions and their parameters

A solution to a linear ordinary differential equation (ODE) $u$ satisfies $\frac{d}{dt}u(t) = \mathbf{M}u(t) + s(t)$, where $\mathbf{M}$ is a linear operator and $s(t)$ is the non-homogeneous term. Assume that the solution to the equation $u(t) = \Phi(t)^\top u$ is a member of a Hilbert space $\mathcal{H}_\kappa$, then $\mathbf{T}u := \left(\frac{d}{dt} - \mathbf{M}(t)\right)\Phi(t)^\top u = s(t)$ for $t \in [t_0, t_1]$. Hence, the differential equation becomes a *linear constraint* for estimating $u$ from samples. In fact, due to differential equations being fully specified by initial conditions, the only unknowns are due to initial conditions. To reveal the linear functional here, one needs to consider the solution to the differential equation, which can be written as $u = \mathbf{T}^\dagger s(t) + \mathbf{C}^\top v$, where $v \in \mathcal{H}_\kappa \backslash \mathrm{span}(\mathbf{T})$ belongs to the null space of $\mathbf{T}$. In this case, we span it with rows of $\mathbf{C}$. Consequently, the unknown element $v$ can be found as $\mathbf{C}(u - \mathbf{T}^\dagger s)$. Thus, what needs to be estimated from samples, is the linear projection $\mathbf{C}u$, since $\mathbf{CT}^\dagger s$ is known a priori. In Appendix F, we discuss implementation and calculation of the operator $\mathbf{C}$ on a discretized domain. Also note that, $\mathbf{Z}$ itself induces a Hilbert space $\mathcal{H}_z$ containing the solutions $u$ (González et al., 2014). However, as we will see, for robust designs it is sometimes convenient to pick a larger Hilbert space $\mathcal{H}_\kappa$ s.t. $\mathcal{H}_z \subset \mathcal{H}_\kappa$ and then apply the differential equation constraint.

**Robust Design** As a specific example, consider an example of a pharmacokinetic model capturing the concentration of a medication in blood and stomach via differential equations: $(d/dt)c_s = -ac_s$ and $(d/dt)c_b = bc_s - dc_b$ for $t \in [t_0, t_1]$, where $c_s$ and $c_b$ are the concentration in stomach and blood respectively. The goal of this analysis is to infer $\gamma = (a, b, d)$ from the measurements of the blood concentration levels with fixed, but perhaps noisy, initial conditions (Gabrielsson and Weiner, 1995). To apply the above procedure, we need a fixed differential operator $\mathbf{T}_\gamma$ to define the operator $\mathbf{C}_\gamma$. However, $\gamma$ is itself unknown. Instead, we can give a generally plausible set of $\gamma \in \Gamma$, initial conditions of concentration in blood $c_b(0) = 0$, and norm constraints on the initial stomach concentration $(c_s(0) - c_{\text{dose}})^2 \leq \lambda^{-1}$ (prior $\mathcal{V}_0$) in order to apply our framework. We use the squared exponential kernel to embed trajectories and consider robust $A$-optimal design (as in Sec. 3.2) by maximizing the worst case metric $\inf_{\gamma \in \Gamma} 1/\mathrm{Tr}(\mathbf{C}_\gamma \mathbf{V}_\lambda^{-1} \mathbf{C}_\gamma^\top)$ with the regularized estimator. This way, the design is appropriate for any $\gamma$. After estimating the trajectory, we can use the estimated trajectories (given $\gamma$) to optimize for $\gamma$ via the maximum likelihood. Fig. 4a presents the concentrations $c_b$ and $c_s$ with their estimates and uncertainties due to the unknown $\gamma$. We show the equally spaced (black) and optimized designs (yellow). The more accurately we can infer the trajectory, the more accurately we can estimate $\gamma$, which we quantitatively show in Appendix E in Fig. 4b, where we decrease the MSE of estimating $\gamma$ by a factor of 10 in comparison to equally spaced design.

### 7.4 Sequential Design: Certifying Lyapunov Stability

Consider a non-linear system with $x \in \mathbb{R}^d$ such that $\frac{d}{dt}x(t) = f(x(t),t) + u(x(t),t) = \mathbf{A}\Phi(x(t),t) + u(x(t),t)$, where the rows of $\mathbf{A}_i \in \mathcal{H}_\kappa$ for $i \in [d]$ model the system dynamics and $\Phi(x(t),t)$ are known evaluation functionals of $\mathcal{H}_\kappa$. We assume that the control laws $u(x(t),t)$ can be written as $u(x(t),t) = \mathbf{B}\Phi(x(t),t)$. We want to understand whether a given a control law $\mathbf{B}$ stabilizes the above system. A common approach is to create an estimate $\hat{\mathbf{A}}$ of $\mathbf{A}$ from data samples of trajectories, and use the controller $\mathbf{B} = \hat{\mathbf{A}} - \mathbf{P}$, where $\mathbf{P}$ is often a simple reverting law with a known gain that compensates the imprecise estimation of $\mathbf{A}$ with $\hat{\mathbf{A}}$, see below for an example. With this choice of $u$, the system can be written as $\frac{d}{dt}x(t) = (\mathbf{A} - \hat{\mathbf{A}} - \mathbf{P})\Phi(x(t),t) = (\tilde{\mathbf{A}} - \mathbf{P})\Phi(x(t),t)$, where the $\tilde{\mathbf{A}} = \mathbf{A} - \hat{\mathbf{A}}$ is the residual error in estimating $\mathbf{A}$. We can certify the stability of the resulting system using a *known* quadratic Lyapunov function as commonly done in control theory, in this case, using $V(x,t) = (x(t) - x_{\text{ref}}(t))^\top \mathbf{\Sigma}(x(t) - x_{\text{ref}}(t))$. Classical stability theory dictates that if the total time derivative of $V$,

$$dV/dt = (x(t) - x_{\text{ref}}(t))^\top \mathbf{\Sigma}(\tilde{\mathbf{A}} - \mathbf{P})\phi(x(t),t) + \partial V/\partial t, \tag{11}$$

is negative for all $x(t) \in O$ then we can guarantee stability in the operating region of $x \in O$ (Khalil, 2002). The above condition defines a linear operator $\mathbf{C}_x = (x - x_{\text{ref}})^\top \mathbf{\Sigma} \cdot \phi(x)$ operating on the unknown $\tilde{\mathbf{A}}$ for each $x \in O$. The linearity is best seen with vectorization as $\theta = \text{vec}(\mathbf{A})$, using the shorthand $z(t) = x(t) - x_{\text{ref}}(t)$, $z^\top \mathbf{\Sigma}\mathbf{A}\phi(x) = \text{vec}(\mathbf{\Sigma}z\phi(x)^\top)^\top \text{vec}(\mathbf{A}) = \mathbf{C}_x\theta$. The operator is parametrized by $x$, $\mathbf{C}_x$ for each $x \in O$. Even for continuous domains $x \in O$, $\mathbf{C}_x$ usually has low-rank structure, which can be calculated and depends on the size $O$ and the size $\mathcal{H}_\kappa$. If the rank of the operator is small (e.g., the operating space is small) then we can certify negativity of Eq. (11) faster than learning the whole $\mathbf{A}$. In other words, we reduce uncertainty only where we need to as in the seminal work of Berkenkamp et al. (2016). We can sequentially query data points from $x$ and check whether $\mathbf{C}_x\theta \leq 0$ for all $x$.

Consider a two dimensional nonlinear system from Lederer et al. (2021),

$$\frac{dx}{dt} = x + \frac{1}{1 + \exp(-2x_1)}\begin{pmatrix}1\\-1\end{pmatrix} + 0.5\begin{pmatrix}\sin(\pi x_2)\\\cos(\pi x_1)\end{pmatrix} + u.$$

The reverting controller is $\mathbf{P}\Phi(x(t),t) = -K(x - x_{\text{ref}}(t)) + (d/dt)x_{\text{ref}}$, where the reference trajectory corresponds to a circle $x_{\text{ref}}(t) = (\sin(t),\cos(t))$. The Lyapunov function is $V = (x(t) - x_{\text{ref}}(t))^\top(x(t) - x_{\text{ref}}(t))$. We assume that we can set the system to an initial condition $x(0)$ and observe a noisy observations of the state $y(0 + \Delta)$ at rapid sampling times $\Delta$. From these, we create a derivative oracle, $(d/dt)x(t) \approx (x(t + \Delta) - x(t))/\Delta$, a common approach in nonlinear data-driven control (Umlauft et al., 2018). We use this example to showcase our adaptive confidence sets. We follow an adaptive stopping rule, where we query new data points if negativity cannot be certified. Due to this adaptive stopping rule, the data is adaptively collected.

We compare our confidence sets (solid) with the classical confidence sets of Abbasi-Yadkori et al. (2011) (dashed) and report upper bounds on the supremum over $x \in O$ of Eq. (11) in Fig. 2c. The operating region is a "tube" around the circular reference trajectory $x_{\text{ref}}(t)$. We see that our confidence sets shrink much faster than the classical confidence sets, since we can eliminate redundant information by projecting onto $\mathbf{C}$. Fig. 2c compares random sampling of datapoints from the whole domain (random) and the operating region (random-ref) and sampling according to uncertainty in the dynamics in the whole domain (unc) and within the tube around the reference trajectory (unc-ref). As expected, focused exploration methods work much better. However, and more importantly, the tightness of our confidence sets enable much quicker stability certification (termination). Note that the classical confidence sets may even *grow* in the cases where redundant information for estimation of $\mathbf{C}$ is inserted into the estimation, e.g., with random sampling.

## 8 CONCLUSION

We considered the problem of learning a linear function of an element in a reproducing kernel Hilbert space. We addressed the challenging case where linear estimators incur a non-negligible bias and provided confidence sets for the two most commonly used linear estimators. We demonstrated the generality of our approach and the tightness of our confidence sets on several challenging applications. We believe our results lay important foundations for principled and efficient experiment design in complex real-world settings.

## Acknowledgement

This project has received funding Swiss National Science Foundation through NFP75 and this publication was created as part of NCCR Catalysis (grant number 180544), a National Centre of Competence in Research funded by the Swiss National Science Foundation.

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
