# Supplementary Material:
# Experimental Design for Linear Functionals in Reproducing Kernel Hilbert Spaces

## A  Estimability results

In the following section, we either describe proof of for implication or equivalences of certain conditions studied in this work. In A.1, we show consequence of Def. 1 which is used in the proofs of confidence sets. In the subsection following it, we establish relationship to classical ODE as in (Pukelsheim, 2006) showing that our bias condition generalizes notion stemming from there.

### A.1  Equivalence of bias conditions

**Proposition 5.** *Let* $\mathbf{L}_\dagger$ *be the interpolation estimator and* $\mathbf{W}_\dagger$ *it associated information matrix then, then*

$$\left\| (\mathbf{C} - \mathbf{L}_\dagger \mathbf{X}) \mathcal{V}_0^{-1/2} \right\|_F^2 \leq \nu^2 \left\| \mathbf{L}_\dagger \right\|_F^2 \implies \left\| \mathbf{W}_\dagger^{1/2} (\mathbf{C} - \mathbf{L}_\dagger \mathbf{X}) \mathcal{V}_0^{-1/2} \right\|_2^2 \leq \nu^2 \tag{12}$$

*Proof.* Note that $\mathbf{V}^\dagger = \mathcal{V}_0^{-1/2} \mathbf{X}^\top \mathbf{K}^{-2} \mathbf{X} \mathcal{V}_0^{-1/2}$, where $\mathbf{K} = \mathbf{X} \mathcal{V}_0^{-1} \mathbf{X}^\top$.

$$
\begin{aligned}
&\quad \text{(LHS)} \\
&= \mathcal{V}_0^{-1/2} (\mathbf{C} - \mathbf{L}_\dagger \mathbf{X})^\top \mathbf{W}_\dagger (\mathbf{C} - \mathbf{L}_\dagger \mathbf{X}) \mathcal{V}_0^{-1/2} \\
&= \mathcal{V}_0^{-1/2} (\mathcal{I} - \mathcal{V}_0^{-1} \mathbf{X}^\top \mathbf{K}^{-1} \mathbf{X})^\top \mathbf{C}^\top (\mathbf{C} \mathcal{V}_0^{-1/2} \mathbf{V}^\dagger \mathcal{V}_0^{-1/2} \mathbf{C}^\top)^{-1} \mathbf{C} (\mathcal{I} - \mathcal{V}_0^{-1} \mathbf{X}^\top \mathbf{K}^{-1} \mathbf{X}) \mathcal{V}_0^{-1/2} \\
&= \mathcal{V}_0^{-1/2} (\mathcal{I} - \mathbf{X} \mathbf{K}^{-1} \mathbf{X}^\top \mathcal{V}_0^{-1}) \mathbf{C}^\top (\mathbf{C} \mathcal{V}_0^{-1/2} \mathbf{V}^\dagger \mathbf{C}^\top)^{-1} \mathcal{V}_0^{-1/2} \mathbf{C} (\mathcal{I} - \mathcal{V}_0^{-1} \mathbf{X}^\top \mathbf{K}^{-1} \mathbf{X}) \mathcal{V}_0^{-1/2} \\
&= (\mathcal{I} - \mathcal{V}_0^{-1/2} \mathbf{X} \mathbf{K}^{-1} \mathbf{X}^\top \mathcal{V}_0^{-1/2}) \mathcal{V}_0^{-1/2} \mathbf{C}^\top (\mathbf{C} \mathcal{V}_0^{-1/2} \mathbf{V}^\dagger \mathcal{V}_0^{-1/2} \mathbf{C}^\top)^{-1} \mathbf{C} \mathcal{V}_0^{-1/2} (\mathcal{I} - \mathcal{V}_0^{-1/2} \mathbf{X}^\top \mathbf{K}^{-1} \mathbf{X} \mathcal{V}_0^{-1/2}) \\
&= (\mathcal{I} - \mathcal{P}) \mathcal{V}_0^{-1/2} \mathbf{C}^\top (\mathbf{C} \mathcal{V}_0^{-1/2} \mathbf{V}^\dagger \mathcal{V}_0^{-1/2} \mathbf{C}^\top)^{-1} \mathbf{C} \mathcal{V}_0^{-1/2} (\mathcal{I} - \mathcal{P})
\end{aligned}
$$

Now let $\tilde{\mathbf{X}} = \mathbf{X} \mathcal{V}_0^{-1/2}$ and $\tilde{\mathbf{C}} = \mathbf{C} \mathcal{V}_0^{-1/2}$. Also notice that, $(\tilde{\mathbf{X}}^\top \tilde{\mathbf{X}})^\dagger = \mathcal{V}_0^{-1/2} \mathbf{X}^\top \mathbf{K}^{-2} \mathbf{X} \mathcal{V}_0^{-1/2} = \mathbf{V}^\dagger$

We can apply Theorem **??**, to get

$$\tilde{\mathbf{C}}^\top (\tilde{\mathbf{C}} (\tilde{\mathbf{X}}^\top \tilde{\mathbf{X}})^\dagger \tilde{\mathbf{C}}^\top)^{-1} \tilde{\mathbf{C}} = \mathcal{V}_0^{-1/2} \mathbf{C}^\top (\mathbf{C} \mathcal{V}_0^{-1/2} (\tilde{\mathbf{X}}^\top \tilde{\mathbf{X}})^\dagger \mathcal{V}_0^{-1/2} \mathbf{C}^\top)^{-1} \mathbf{C} \mathcal{V}_0^{-1/2} \preceq c \tilde{\mathbf{X}}^\top \tilde{\mathbf{X}} + \mathcal{I} \nu^2$$

Using the above result, we show that

$$
\begin{aligned}
\text{(LHS)} \quad &\preceq \quad (\mathcal{I} - \mathcal{P})(\mathcal{V}_0^{-1/2} \mathbf{X}^\top \mathbf{X} \mathcal{V}_0^{-1/2} + \mathcal{I} \nu^2)(\mathcal{I} - \mathcal{P}) \\
&= \quad \nu^2 (\mathcal{I} - \mathcal{P})^2 \preceq \mathcal{I} \nu^2
\end{aligned}
$$

where we have used the fact that $\mathcal{I} - \mathcal{P}$ is projection matrix with orthogonal span to $\mathcal{V}_0^{-1/2} \mathbf{X}^\top \mathbf{X} \mathcal{V}_0^{-1/2}$.

$\square$

**Proposition 6.** *Bias* $\left\| (\mathbf{C} - \mathbf{L}_\dagger \mathbf{X}) \mathcal{V}_0^{-1/2} \right\|$ *depends only on its support not the allocations.*

*Proof.*

Let $\mathbf{L}_\eta$ be the estimator with design $\eta$, where

$$
\begin{aligned}
\mathbf{L}_\eta \quad &= \quad \mathbf{C} \mathcal{V}_0^{-1} \mathbf{X}^\top \mathbf{D}(\eta)^{1/2} (\mathbf{D}(\eta)^{1/2} \mathbf{X} \mathcal{V}_0^{-1} \mathbf{X}^\top \mathbf{D}(\eta)^{1/2})^{-1} \tag{13} \\
&= \quad \mathbf{C} \mathcal{V}_0^{-1} \mathbf{X} (\mathbf{X} \mathcal{V}_0^{-1} \mathbf{X}^\top)^{-1} \mathbf{D}(\eta)^{-1/2} = \mathbf{L}_\dagger \mathbf{D}(\eta)^{-1/2} \tag{14}
\end{aligned}
$$

But at the same time $\mathbf{D}(\eta)^{1/2} \mathbf{X}$, hence these two cancel each other.

$\square$

## A.2 Basic equivalences and relations

To relate the estimability to terms used in the ED literature, we state the definition of the *feasibility cone* as in Pukelsheim (2006). We show that our condition in Def. 1 and Pukelsheims and estimability are equivalent under classical assumptions. We use a different formulation of the relative-bias condition due to Proposition 5.

**Definition 3** ((Pukelsheim, 2006)). Let $\mathbf{C} : \mathcal{H}_\kappa \to \mathbb{R}^k$, denote $\mathcal{A}(\mathbf{C}) = \{\mathbf{A} \in \mathcal{P}(\mathcal{H}_\kappa) :$ range$(\mathbf{C}^\top) \subseteq$ range$(\mathbf{A})\}$, where $\mathcal{P}(\mathcal{H}_\kappa)$ is the space of positive-definite operators on $\mathcal{H}_\kappa$. We call $\mathcal{A}(\mathbf{C})$ the feasiblity cone of $\mathbf{K}$.

This definition is sometimes used as restatement of the estimability property.

**Lemma 1** (Equivalence in $\nu = 0$). *Let $\mathbf{C} \in \mathbb{R}^{k \times d}$ full rank $k$. Let $\mathbf{X} \in \mathbb{R}^{n \times d}$, then if $\mathbf{M} = \mathbf{X}^\top \mathbf{X}$, the following are equivalent,*

1. ***Estimability:** There exists $\mathbf{L} \in \mathbb{R}^{k \times n}$ s.t. $\mathbf{C} = \mathbf{L}\mathbf{X}$.*

2. ***Feasibility cone:** $\mathbf{M} \in \mathcal{A}(\mathbf{C})$*

*Proof.*

- (2) $\implies$ (1): There exists $\mathbf{B}$ s.t. $\mathbf{C} = \mathbf{B}\mathbf{M}$, and $\mathbf{C} = \mathbf{B}\mathbf{X}^\top\mathbf{X}$, hence we can define $\mathbf{L} = \mathbf{B}\mathbf{X}$.

- (1) $\implies$ (2): As such $\mathbf{C} = \mathbf{L}\mathbf{X}$ implies that range$(\mathbf{C}) \subseteq$ range$(\mathbf{X}^\top) =$ range$(\mathbf{X}^\top\mathbf{X})$.

$\square$

**Definition 4** (Projected data). Let $z(x) \in \mathbb{R}^p$ be a vector field s.t. $\Phi(x)\mathcal{V}_0^{-1/2} = z(x)\mathbf{C}\mathcal{V}_0^{-1/2} + j(x)$, where $x \in \mathcal{S} \subseteq \mathcal{X}$ and $j(x) \in \mathcal{H}_\kappa$, $|\mathcal{S}| = n$, such that $\mathbf{C}\mathcal{V}_0^{-1/2}j(x) = 0$. We call this vector field *projected data*. If in addition if $\{z(x) : x \in \mathcal{S}\}$ spans $\mathbb{R}^p$ it is said to be *approximately low-rank*.

**Lemma 2.** *The assumption in Definition 4 implies the assumption in Definition 1 with $\nu = \|\mathbf{J}\|_k \frac{\|\mathbf{Z}^\dagger\|_F}{\|\mathbf{L}_\dagger\|_F}$.*

*Proof.* Since $\mathbf{Z}$ spans whole $\mathbb{R}^k$, there exists a unique left pseudo-inverse. $\mathbf{Z}^\dagger\mathbf{X}\mathcal{V}_0^{-1/2} = \mathbf{C}\mathcal{V}_0^{-1/2} + \mathbf{Z}^\dagger\mathbf{J}$.

$$(\mathbf{C} - \mathbf{Z}^\dagger\mathbf{X})\mathcal{V}_0^{-1/2} = \mathbf{Z}^\dagger\mathbf{J}$$

Now taking the Frobenius norm,

$$\left\|(\mathbf{C} - \mathbf{Z}^\dagger\mathbf{X})\mathcal{V}_0^{-1/2}\right\|_F \leq \left\|\mathbf{Z}^\dagger\mathbf{J}\right\|_F = \|\mathbf{J}\|_F \left\|\mathbf{Z}^\dagger\right\|_F$$

So,

$$\frac{\left\|(\mathbf{C} - \mathbf{Z}^\dagger\mathbf{X})\mathcal{V}_0^{-1/2}\right\|_F}{\|\mathbf{Z}^\dagger\|_F} \leq \|\mathbf{J}\|_F$$

Now, since pseudo-inverse minimizes the LHS, we know that,

$$\left\|(\mathbf{C} - \mathbf{L}_\dagger\mathbf{X})\mathcal{V}_0^{-1/2}\right\|_F \leq \left\|(\mathbf{C} - \mathbf{Z}^\dagger\mathbf{X})\mathcal{V}_0^{-1/2}\right\|_F \leq \|\mathbf{J}\|_k \frac{\|\mathbf{Z}^\dagger\|_F}{\|\mathbf{L}_\dagger\|_F}\|\mathbf{L}_\dagger\|_F$$

$\square$

# B Confidence Sets: Proofs

This section includes proofs for the concentration results presented in the main text. In Section, B.4 we restate and prove results from Mutný et al. (2020) in the current notation for easier comparison.

## B.1 Fixed Design with Interpolator

The following Theorem tries to qualify whether a design contains sufficient information to reduce confidence on a specific subspace $\mathbf{C}$ to the desired error.

**Theorem 7** (Interpolation (Fixed Design)). *Under regression model in Eq.* (1) *with $T$ data point evaluations, let $\hat\theta$ be the estimate as in* (3). *Further let $\mathbf{X}$ be s.t. satisfy the approximate estimability condition of Def.* 1 *with $T \geq k$. Then,*

$$\mathrm{P}\left(\left\|\mathbf{C}(\hat\theta - \theta)\right\|_{\mathbf{W}} \geq \sigma\sqrt{\xi(\delta)} + \frac{\nu}{\sqrt{\lambda}}\right) \leq \delta, \tag{15}$$

*where* $\mathbf{W} = (\mathbf{C}\mathcal{V}_0^{-1/2}\mathbf{V}^\dagger\mathcal{V}_0^{-1/2}\mathbf{C}^\top)^{-1}$, $\mathbf{V}^\dagger = \mathcal{V}_0^{-1/2}\mathbf{X}^\top\mathbf{K}^{-2}\mathbf{X}\mathcal{V}_0^{-1/2}$, *and* $\xi(\delta) = p + 2\left(\sqrt{p\log(\frac{1}{\delta})} + \log\left(\frac{1}{\delta}\right)\right)$.

*Proof.* The pseudo inverse estimate of $\mathbf{C}\theta$ is $\mathbf{C}\hat\theta = \mathbf{C}\mathcal{V}_0^{-1}\mathbf{X}^\top(\mathbf{X}\mathcal{V}_0^{-1}\mathbf{X}^\top)^{-1}y$, where $\mathbf{X} \in \mathbb{R}^{T\times m}$.

$$
\begin{aligned}
\left\|\mathbf{C}(\hat\theta - \theta)\right\|_{\mathbf{W}}^2 &= \left\|\mathbf{C}(\mathcal{V}_0^{-1}\mathbf{X}^\top(\mathbf{X}\mathcal{V}_0^{-1}\mathbf{X}^\top)^{-1}(\mathbf{X}\theta + \epsilon) - \theta)\right\|_{\mathbf{W}}^2 \\
&= \left\|\mathbf{C}(\mathcal{V}_0^{-1}\mathbf{X}^\top(\mathbf{X}\mathcal{V}_0^{-1}\mathbf{X}^\top)^{-1}\mathbf{X} - \mathcal{I})\theta + \mathbf{C}\mathcal{V}_0^{-1}(\mathbf{X}^\top(\mathbf{X}\mathcal{V}_0^{-1}\mathbf{X}^\top)^{-1}\epsilon)\right\|_{\mathbf{W}}^2 \\
&\leq \left\|\mathbf{C}(\mathcal{V}_0^{-1}\mathbf{X}^\top(\mathbf{X}\mathcal{V}_0^{-1}\mathbf{X}^\top)^{-1}\mathbf{X} - \mathcal{I})\mathcal{V}_0^{-1/2}\mathcal{V}_0^{1/2}\theta\right\|_{\mathbf{W}}^2 \\
&\quad + \left\|\mathbf{C}\mathcal{V}_0^{-1}\mathbf{X}^\top(\mathbf{X}\mathcal{V}_0^{-1}\mathbf{X}^\top)^{-1}\epsilon\right\|_{\mathbf{W}}^2 \\
&\leq \left\|\mathbf{W}^{1/2}\mathbf{C}(\mathcal{V}_0^{-1}\mathbf{X}^\top(\mathbf{X}\mathcal{V}_0^{-1}\mathbf{X}^\top)^{-1}\mathbf{X} - \mathcal{I})\mathcal{V}_0^{-1/2}\right\|_2^2 \|\theta\|_{\mathcal{V}_0} \\
&\quad + \left\|\mathbf{C}\mathcal{V}_0^{-1}\mathbf{X}^\top(\mathbf{X}\mathcal{V}_0^{-1}\mathbf{X}^\top)^{-1}\epsilon\right\|_{\mathbf{W}}^2 \\
&= \lambda^{-1}\nu^2 + \left\|\mathbf{C}\mathcal{V}_0^{-1}\mathbf{X}^\top(\mathbf{X}\mathcal{V}_0^{-1}\mathbf{X}^\top)^{-1}\epsilon\right\|_{\mathbf{W}}^2
\end{aligned}
$$

where the second to last line we used the relative-bias assumption, according to the Proposition 5, where we use the Def. 1 and $T \geq k$. The operator $\mathcal{I}$ is identify operator on $\mathcal{H}_k$.

If $\epsilon_G$ were Gaussially distributed then, $\mathbf{C}\mathcal{V}_0^{-1}\mathbf{X}^\top(\mathbf{X}\mathcal{V}_0^{-1}\mathbf{X}^\top)^{-1}\epsilon_G$ is distributed as $\mathcal{N}(0, \mathbf{C}\mathbf{V}^\dagger\mathbf{C}^\top)$. Likewise, $q_G = \mathbf{W}^{1/2}\mathbf{C}\mathcal{V}_0^{-1/2}\mathbf{X}^\top(\mathbf{X}\mathcal{V}_0^{-1}\mathbf{X}^\top)^{-1}\epsilon_G$ is distributed as $\mathcal{N}(0, \mathbf{I}_k)$. Since $\epsilon$ is sub-Gaussian $q$ needs to have tails of distribution which are below the tails of $k$-dimensional gaussian. In other words $P(|q_i| \leq t) \leq P(|q_{G_i}| \leq t)$. As such, $P(q_i^2 \leq t^2) \leq P(|q_{G_i}|^2 \leq t)$, where $q_{G_i}^2$ is chi-squared distributed. Using concentration for chi-squared random variables, $\mathrm{P}\left(\|z\|_2^2 - \sigma^2 p \geq \sigma^2(2px + 2\sqrt{px})\right) \leq \exp(-x)$ as in Laurent and Massart (2000). Rearranging the expression leads to the result with $\xi(\delta) = (p + 2\log(\frac{1}{\delta}) + 2\sqrt{p\log(\frac{1}{\delta})})$, multiplied by $\sigma^2$.

The last final inequality in the statement of the probability follows from taking a square root and triangle inequality, which finishes the proof. $\square$

## B.2 Fixed Design with Regularized Estimator

**Proposition 8** (Fixed Design Ridge Regression). *Under regression model in Eq.* (1) *with $T$ data point evaluations, let $\hat\theta_\lambda$ be the regularized estimate as in* (5). *Then,*

$$\mathrm{P}\left(\left\|\mathbf{C}(\hat\theta_\lambda - \theta)\right\|_{\mathbf{W}_\lambda} \geq \sqrt{\beta(\delta)} = \sqrt{\xi(\delta)} + 1\right) \leq \delta,$$

*where* $\mathbf{W}_\lambda = \sigma^{-2}(\mathbf{C}\mathcal{V}_\lambda^{-1}\mathbf{C}^\top)^{-1}$, $\mathcal{V}_\lambda^{-1} = (\mathbf{X}^\top\mathbf{X} + \lambda\sigma^2\mathcal{V}_0)^{-1}$. *In addition,* $\xi(\delta) = p + 2\left(\sqrt{p\log(\frac{1}{\delta})} + \log\left(\frac{1}{\delta}\right)\right)$.

*Proof.* Notice that invertibility of $\mathbf{W}$ is guaranteed by full rank $\mathbf{C}$ and invertibility of $\mathbf{V}_\lambda$. Using shorthand $\mathcal{V} = \mathbf{X}^\top\mathbf{X}$,

$$
\begin{aligned}
\left\|\mathbf{C}(\hat{\theta}_\lambda - \theta)\right\|_{\mathbf{W}_\lambda}^2 &= \left\|\mathbf{C}\mathcal{V}_0^{-1}\mathbf{X}^\top(\mathbf{X}\mathcal{V}_0^{-1}\mathbf{X}^\top + \sigma^2\lambda\mathbf{I})^{-1}(\mathbf{X}\theta + \epsilon) - \theta)\right\|_{\mathbf{W}_\lambda}^2 \\
&= \left\|\mathbf{C}\mathcal{V}_0^{-1/2}(\mathcal{V}_0^{-1/2}\mathbf{X}^\top\mathbf{X}\mathcal{V}_0^{-1/2} + \sigma^2\lambda\mathbf{I})^{-1}\mathcal{V}_0^{-1/2}\mathbf{X}^\top(\mathbf{X}\theta + \epsilon) - \theta)\right\|_{\mathbf{W}_\lambda}^2 \\
&= \left\|\mathbf{C}\mathcal{V}_\lambda^{-1}\mathbf{X}^\top(\mathbf{X}\theta + \epsilon) - \theta)\right\|_{\mathbf{W}_\lambda}^2 \\
&= \left\|\mathbf{C}(\mathcal{V}_\lambda^{-1}\mathbf{X}^\top\epsilon + \mathcal{V}_\lambda^{-1}\mathcal{V}\theta - \theta)\right\|_{\mathbf{W}_\lambda}^2 \\
&\leq \left\|\mathbf{C}\mathcal{V}_\lambda^{-1}\mathbf{X}^\top\epsilon\right\|_{\mathbf{W}_\lambda}^2 + \left\|\mathbf{C}(\mathcal{V}_\lambda^{-1}\mathbf{X}^\top\mathbf{X} - \mathcal{I})\theta\right\|_{\mathbf{W}_\lambda}^2
\end{aligned}
$$

The second term,

$$
\left\|\mathbf{C}\mathcal{V}_\lambda^{-1}(\mathcal{V}_0\lambda\sigma^2)\theta)\right\|_{\mathbf{W}_\lambda}^2 \overset{(64)}{\leq} \sigma^2\left\|\mathcal{V}_\lambda^{-1}(\mathcal{V}_0\lambda)\theta)\right\|_{\mathcal{V}_\lambda}^2 \tag{16}
$$

$$
= \lambda\sigma^2\theta^\top\mathcal{V}_0\mathcal{V}_\lambda^{-1}(\mathcal{V}_0\lambda)\theta) \leq \lambda\theta^\top\mathcal{V}_0\theta \leq 1. \tag{17}
$$

Let us analyze the first term,

$$
\left\|\mathbf{C}\mathcal{V}_\lambda^{-1}\mathbf{X}^\top\epsilon\right\|_{\mathbf{W}_\lambda}^2 = \epsilon^\top\mathbf{X}\mathcal{V}_\lambda^{-1}\mathbf{C}^\top\mathbf{W}_\lambda\mathbf{C}\mathcal{V}_\lambda^{-1}\mathbf{X}^\top\epsilon \tag{18}
$$

The distribution of $\mathbf{C}\mathcal{V}_\lambda^{-1}\mathbf{X}^\top\epsilon$ is $\mathcal{N}(0, \sigma^2\mathbf{C}\mathcal{V}_\lambda^{-1}\mathbf{V}\mathcal{V}_\lambda^{-1}\mathbf{C}^\top)$. Further, $(\sigma\mathbf{C}\mathcal{V}_\lambda^{-1}\mathbf{C}^\top)^{-1/2}\mathbf{C}\mathcal{V}_\lambda^{-1}\mathbf{X}^\top\epsilon$ is distributed as $z \sim \mathcal{N}\left(0, (\mathbf{C}\mathcal{V}_\lambda^{-1}\mathbf{C}^\top)^{-1/2}\mathbf{C}\mathcal{V}_\lambda^{-1}\mathcal{V}\mathcal{V}_\lambda^{-1}\mathbf{C}^\top(\mathbf{C}\mathcal{V}_\lambda^{-1}\mathbf{C}^\top)^{-1/2}\right)$. Let us call the covariance matrix $\boldsymbol{\Sigma}$. It is easy to see that $\boldsymbol{\Sigma} \preceq \mathbf{P}$, an projection matrix with $k$ unit eigenvalues.The random variable $z$ can be generated as $z = q^\top\boldsymbol{\Sigma}q$, where $q \sim \mathcal{N}(0, \mathbf{I}_n)$. Our goal is to bound,$P(\|z\|_2^2 \geq t)$. Using eigenvalue decomposition of $\boldsymbol{\Sigma} = \mathbf{Q}\boldsymbol{\Lambda}\mathbf{Q}^\top$, we can show

$$
\mathrm{P}(\|z\|_2^2 \geq t) = \mathrm{P}(q^\top\mathbf{Q}\boldsymbol{\Lambda}\mathbf{Q}^\top q^\top \geq t) = \mathrm{P}(q^\top\boldsymbol{\Lambda}q^\top \geq t) \leq \mathrm{P}(q^\top\mathbf{P}q^\top \geq t) = \mathrm{P}(\|\tilde{q}\|_2^2 \geq t)
$$

where $\tilde{q}$ is $p$-dimensional standard normal. Using concentration for chi-squared random variables, $\mathrm{P}\left(\|z\|_2^2 - \sigma^2 p \geq \sigma^2(2px + 2\sqrt{px})\right) \leq \exp(-x)$ as in [Laurent and Massart (2000)](). Rearranging the expression leads to the result with $\xi(\delta) = (p + 2\log(\frac{1}{\delta}) + 2\sqrt{p\log(\frac{1}{\delta})})$, multiplied by $\sigma^2$. To deal with sub-Gaussianity we apply the same trick as in the previous Theorem.

$\square$

## B.3 Adaptive design and Regularized Estimator

**Theorem 9** (Adaptive Design Regularized Regression). *Under the regression model in Eq.* (1) *with $t$ adaptively collected data points, let $\hat{\theta}_{\lambda,t}$ be the regularized estimate as in* (5). *Further, assume that $\mathbf{Z}$ is as in Def.* 2 *where $\mathbf{X}_t = \mathbf{Z}_t\mathbf{C} + \mathbf{J}_t\mathcal{V}_0^{1/2}$. Then for all $t \geq 0$,*

$$
\left\|\mathbf{C}(\hat{\theta}_t - \theta)\right\|_{\boldsymbol{\Omega}_{\lambda,t}} \leq \sqrt{2\log\left(\frac{1}{\delta}\frac{\det(\boldsymbol{\Omega}_{\lambda,t})^{1/2}}{\det(\lambda\mathbf{S})^{1/2}}\right) + 1} \tag{19}
$$

*with probability $1 - \delta$, where $\boldsymbol{\Omega}_{\lambda,t} = \frac{1}{\sigma^2}\mathbf{Z}_t^\top\mathbf{Z}_t + \lambda\mathbf{S}$ and $\mathbf{S} = (\mathbf{C}\mathcal{V}_0^{-1}\mathbf{C}^\top)^{-1}$.*

*Proof.* Notice that the theorem is stated with $\mathbf{S}$ and $\mathbf{S}^{-1}$ having inverted roles in the proof, however the result above follows by swapping the two.

Note that $\mathbf{J}\mathcal{V}_0^{-1/2}\mathbf{C}^\top = 0$ as well as $\mathbf{C}\mathcal{V}_0^{-1/2}\mathbf{J}^\top = 0$ due to the assumption. Also, notice that $\mathbf{X} = \mathbf{Z}\mathbf{C} + \mathbf{J}\mathcal{V}_0^{1/2}$. We drop the $t$ subscript for brevity.

$$\left\|\mathbf{C}\hat{\theta} - \mathbf{C}\theta\right\|_{\mathbf{\Omega}_\lambda}^2 \;=\; \left\|\mathbf{C}(\mathcal{V}_\lambda^{-1}\mathbf{X}^\top(\mathbf{X}\theta + \epsilon) - \theta)\right\|_{\mathbf{\Omega}_\lambda}^2 = \left\|\mathbf{C}(\mathcal{V}_\lambda^{-1}\mathbf{X}^\top\epsilon + \mathcal{V}_\lambda^{-1}(\mathcal{V} - \mathcal{V}_\lambda)\theta)\right\|_{\mathbf{\Omega}_\lambda}^2$$

$$\leq \; \left\|\mathbf{C}\mathcal{V}_\lambda^{-1}\mathbf{X}^\top\epsilon\right\|_{\mathbf{\Omega}_\lambda}^2 + \left\|\mathbf{C}\mathcal{V}_\lambda^{-1}(\mathcal{V} - \mathcal{V}_\lambda)\theta)\right\|_{\mathbf{\Omega}_\lambda}^2$$

Now we analyze the two terms separately, The first term,

$$\left\|\mathbf{C}\mathcal{V}_\lambda^{-1}\mathbf{X}^\top\epsilon\right\|_{\mathbf{\Omega}_\lambda}^2 \;=\; \left\|\mathbf{C}\mathcal{V}_0^{-1}\mathbf{X}^\top(\mathbf{X}\mathcal{V}_0^{-1}\mathbf{X}^\top + \sigma^2\lambda\mathbf{I})^{-1}\epsilon\right\|_{\mathbf{\Omega}_\lambda}^2$$

$$=\; \left\|\mathbf{C}\mathcal{V}_0^{-1}(\mathbf{Z}\mathbf{C} + \mathbf{J}\mathcal{V}_0^{1/2})^\top(\mathbf{X}\mathcal{V}_0^{-1}\mathbf{X}^\top + \sigma^2\lambda\mathbf{I})^{-1}\epsilon\right\|_{\mathbf{\Omega}_\lambda}^2$$

$$\leq\; \left\|\mathbf{C}\mathcal{V}_0^{-1}(\mathbf{Z}\mathbf{C})^\top(\mathbf{X}\mathcal{V}_0^{-1}\mathbf{X}^\top + \sigma^2\lambda\mathbf{I})^{-1}\epsilon\right\|_{\mathbf{\Omega}_\lambda}^2$$

$$+\; \Big\|\underbrace{\mathbf{C}\mathcal{V}_0^{-1/2}\mathbf{J}^\top}_{=0}(\mathbf{X}\mathcal{V}_0^{-1}\mathbf{X}^\top + \sigma^2\lambda\mathbf{I})^{-1}\epsilon\Big\|_{\mathbf{\Omega}_\lambda}^2$$

$$=\; \left\|\mathbf{C}\mathcal{V}_0^{-1}\mathbf{C}^\top\mathbf{Z}^\top((\mathbf{Z}\mathbf{C} + \mathbf{J}\mathcal{V}_0^{1/2})\mathcal{V}_0^{-1}(\mathbf{Z}\mathbf{C} + \mathbf{J}\mathcal{V}_0^{1/2})^\top + \sigma^2\lambda\mathbf{I})^{-1}\epsilon\right\|_{\mathbf{\Omega}_\lambda}^2$$

$$=\; \Big\|\mathbf{C}\mathcal{V}_0^{-1}\mathbf{C}^\top\mathbf{Z}^\top(\mathbf{Z}\mathbf{C}\mathcal{V}_0^{-1}\mathbf{C}^\top\mathbf{Z} + \underbrace{\mathbf{J}\mathcal{V}_0^{-1/2}\mathbf{C}\,\mathbf{Z}^\top}_{=0} + \mathbf{Z}\underbrace{\mathbf{C}\mathcal{V}_0^{-1/2}\mathbf{J}^\top}_{=0} + \mathbf{J}\mathbf{J}^\top\sigma^2\lambda\mathbf{I})^{-1}\epsilon\Big\|_{\mathbf{\Omega}_\lambda}^2$$

$$=\; \left\|\mathbf{C}\mathcal{V}_0^{-1}\mathbf{C}^\top\mathbf{Z}^\top(\mathbf{Z}\mathbf{C}\mathcal{V}_0^{-1}\mathbf{C}^\top\mathbf{Z} + \mathbf{J}\mathbf{J}^\top\sigma^2\lambda\mathbf{I})^{-1}\epsilon\right\|_{\mathbf{\Omega}_\lambda}^2$$

$$\leq\; \left\|\mathbf{C}\mathcal{V}_0^{-1}\mathbf{C}^\top\mathbf{Z}^\top(\mathbf{Z}\mathbf{C}\mathcal{V}_0^{-1}\mathbf{C}^\top\mathbf{Z} + \sigma^2\lambda\mathbf{I})^{-1}\epsilon\right\|_{\mathbf{\Omega}_\lambda}^2$$

Let us define shorthand $\mathbf{S}^{-1} = \mathbf{C}\mathcal{V}_0^{-1}\mathbf{C}^\top$ which is $\mathbb{R}^{k\times k}$ p.s.d. matrix.

$$\left\|\mathbf{C}\mathcal{V}_\lambda^{-1}\mathbf{Z}^\top\epsilon\right\|_{\mathbf{\Omega}_\lambda}^2 \;\leq\; \left\|\mathbf{S}^{-1/2}\mathbf{S}^{-1/2}\mathbf{Z}^\top(\mathbf{Z}\mathbf{S}^{-1/2}\mathbf{S}^{-1/2}\mathbf{Z} + \sigma^2\lambda\mathbf{I})^{-1}\epsilon\right\|_{\mathbf{\Omega}_\lambda}^2 \tag{20}$$

$$=\; \left\|\mathbf{S}^{-1/2}(\mathbf{S}^{-1/2}\mathbf{Z}^\top\mathbf{Z}\mathbf{S}^{-1/2} + \sigma^2\lambda\mathbf{I})^{-1}\mathbf{S}^{-1/2}\mathbf{Z}^\top\epsilon\right\|_{\mathbf{\Omega}_\lambda}^2 \tag{21}$$

$$=\; \left\|(\mathbf{Z}^\top\mathbf{Z} + \sigma^2\lambda\mathbf{S})^{-1}\mathbf{Z}^\top\epsilon\right\|_{\mathbf{\Omega}_\lambda}^2 \tag{22}$$

$$=\; \left\|\mathbf{Z}^\top\epsilon\right\|_{\mathbf{\Omega}_\lambda^{-1}}^2 \tag{23}$$

The term above is so called self-normalized noise, which can be handled by techniques of de la Peña et al. (2009) popularized by Abbasi-Yadkori et al. (2011). From now on the proof is generic. Let us define, the noise process $S_t = \sum_{i=1}^t z_i \frac{\epsilon_i}{\sigma^2}$ and variance $\mathbf{V}_t = \frac{z_i z_i^\top}{\sigma^2}$. Also, let $M_t(x) = \exp(S_t^\top x - \frac{1}{2}x^\top\mathbf{V}_t x)$ be a process with index $t$.

Due to sub-gaussianity of $\epsilon_i$, $M_t(x)$ is a super-martingale under filtration that includes all $x_{t-1}, \epsilon_{t-1}, \ldots x_1, \epsilon_1$. Now it is worth noting that this all has been introduces since $\exp(\left\|\mathbf{Z}^\top\epsilon\right\|_{\mathbf{\Omega}_\lambda^{-1}}^2) = \sup_x M_t(x)$. This relation allow us to see that if we can upper bound $\sup_x M_t(x)$ we get what we want. We proceed by pseudo-maximization. We pick a fixed probability distribution $h(x)$ and define a process $\bar{M}_t = \int M_t(x)h(x)dx$. Since, $h(x)$ is fixed and normalized $\bar{M}_t$ is also a super-martingale. It turn out that $h(x) \sim \mathcal{N}(0, \lambda^{-1}\mathbf{S}^{-1})$ will achieve the desired result.

$$\bar{M}_t \;=\; \int M_t(x)h(x)dx = \frac{1}{\sqrt{(2\pi)^k \det((\lambda)^{-1}\mathbf{S}^{-1})}} \int_{\mathbb{R}^k} \exp(x^\top S_t - \frac{1}{2}\left\|x\right\|_{\mathbf{V}_t} - \frac{\lambda}{2}\left\|x\right\|_{\mathbf{S}})dx$$

$$=\; \left(\frac{\det(\mathbf{\Omega}_{\lambda,t})}{\det(\mathbf{S})}\right)^{1/2} \exp(\frac{1}{2}\left\|S_t\right\|_{\mathbf{\Omega}_{\lambda,t}}^2)$$

where we have applied standard rules for Gaussian integrals. Now, we use Ville's martingale inequality to bound,

$$P\left(\sup \bar{M}_t \geq \frac{1}{\alpha}\right) \quad \leq \quad \mathbb{E}[\bar{M}_t]\alpha = \alpha \qquad (24)$$

$$P\left(\log(\sup \bar{M}_t) \geq \log\left(\frac{1}{\alpha}\right)\right) \quad \leq \quad \alpha \qquad (25)$$

$$P\left(\|S_t\|^2_{\mathbf{\Omega}^{-1}_{\lambda,t}} \geq 2\log\left(\frac{1}{\alpha}\right) + 2\log\left(\left(\frac{\det(\mathbf{S})}{\det(\mathbf{\Omega}_{\lambda,t})}\right)^{-1/2}\right)\right) \quad \leq \quad \alpha \qquad (26)$$

$$P(\|S_t\|^2_{\mathbf{\Omega}^{-1}_{\lambda,t}} \geq 2\log\left(\frac{1}{\alpha}\det\left(\frac{\det(\mathbf{S})}{\det(\mathbf{\Omega}_{\lambda,t})}\right)^{-1/2}\right) \quad \leq \quad \alpha \qquad (27)$$

$$(28)$$

which finishes bounding the first term by $2\log\left(\frac{1}{\alpha}\left(\frac{\det(\mathbf{\Omega}_{\lambda,t})}{\det(\mathbf{S})}\right)^{1/2}\right)$

Now we turn to the second term,

$$\left\|\mathbf{C}\mathcal{V}^{-1}_\lambda(\mathcal{V} - \mathcal{V}_\lambda)\theta)\right\|^2_{\mathbf{\Omega}_\lambda} \quad = \quad \left\|\mathbf{C}\mathcal{V}^{-1}_\lambda(\mathcal{V}_0\lambda\sigma^2)\theta)\right\|^2_{\mathbf{\Omega}_\lambda} \qquad (29)$$

$$\overset{(34)}{\leq} \quad \left\|\mathbf{C}\mathcal{V}^{-1}_\lambda(\mathcal{V}_0\lambda\sigma^2)\theta)\right\|^2_{\mathbf{W}_\lambda} \qquad (30)$$

$$\overset{(64)}{\leq} \quad \left\|\mathcal{V}^{-1}_\lambda(\mathcal{V}_0\lambda)\theta)\right\|^2_{\mathcal{V}_\lambda} \qquad (31)$$

$$= \quad \lambda\theta^\top\mathcal{V}_0\mathcal{V}^{-1}_\lambda(\mathcal{V}_0\lambda)\theta) \leq \lambda^2\theta^\top\mathcal{V}_0\theta \leq 1 \qquad (32)$$

$$(33)$$

This proves the result. $\qquad\square$

**Lemma 3** (Ordering).

$$\mathbf{W}_\lambda \preceq \mathbf{\Omega}_\lambda \qquad (34)$$

*Proof.* Consider series of psd. manipulations,

$$\mathbf{C}^\top\mathbf{W}_\lambda\mathbf{C} \quad = \quad \sigma^{-2}\mathbf{C}^\top(\mathbf{C}(\mathbf{X}^\top\mathbf{X} + \sigma^2\lambda\mathcal{V}_0)^{-1}\mathbf{C}^\top)^{-1}\mathbf{C}$$

$$\overset{(64)}{\preceq} \quad \sigma^{-2}\mathbf{X}^\top\mathbf{X} + \lambda\mathcal{V}_0$$

$$= \quad \sigma^{-2}(\mathbf{C}^\top\mathbf{Z}^\top\mathbf{Z}\mathbf{C} + \mathbf{C}^\top\mathbf{Z}^\top\mathbf{J}\mathcal{V}^{1/2}_0 + \mathcal{V}^{1/2}_0\mathbf{J}^\top\mathbf{Z}\mathbf{C} + \mathcal{V}^{1/2}_0\mathbf{J}^\top\mathbf{J}\mathcal{V}^{1/2}_0)$$

$$\quad + \lambda\mathcal{V}_0$$

$$\mathbf{C}\mathcal{V}^{-1}_0\mathbf{C}^\top\mathbf{W}_\lambda\mathbf{C}\mathcal{V}^{-1}_0\mathbf{C}^\top \quad \preceq \quad \mathbf{C}\mathcal{V}^{-1}_0(\sigma^{-2}\mathbf{C}^\top\mathbf{Z}^\top\mathbf{Z}\mathbf{C} + \sigma^{-2}\mathbf{C}^\top\mathbf{Z}^\top\mathbf{J}\mathcal{V}^{1/2}_0$$

$$\quad + \sigma^{-2}\mathcal{V}^{1/2}_0\mathbf{J}^\top\mathbf{Z}\mathbf{C} + \sigma^{-2}\mathcal{V}^{1/2}_0\mathbf{J}^\top\mathbf{J}\mathcal{V}^{1/2}_0 + \lambda\mathcal{V}_0)\mathcal{V}^{-1}_0\mathbf{C}^\top$$

$$= \quad \sigma^{-2}\mathbf{S}^{-1}\mathbf{Z}^\top\mathbf{Z}\mathbf{S}^{-1} + \lambda\mathbf{C}\mathcal{V}^{-1}_0\mathbf{C}^\top$$

$$= \quad \sigma^{-2}\mathbf{S}^{-1}\mathbf{Z}^\top\mathbf{Z}\mathbf{S}^{-1} + \lambda\mathbf{S}^{-1}$$

$$\mathbf{W}_\lambda \quad \preceq \quad \frac{\mathbf{Z}^\top\mathbf{Z}}{\sigma^2} + \lambda\mathbf{S} = \mathbf{\Omega}_\lambda$$

where we use the fact that $\mathbf{C}\mathcal{V}^{-1/2}_0\mathbf{J}^\top = 0$. $\qquad\square$

**Lemma 4** (confidence parameter size). *Under assumptions of Thm. 3, where* $\|\Phi(x)\|^2_k \leq L^2$, $\mathcal{V}_0 = \mathcal{I}$,

$$\sqrt{2\log\left(\frac{1}{\delta}\frac{\det(\mathbf{\Omega}_{\lambda,t})^{1/2}}{\det(\lambda\mathbf{S})^{1/2}}\right)} \leq \sqrt{p\log\left(\frac{tL^2}{p\lambda} + 1\right) + 2\log(1/\delta)} = \mathcal{O}(\sqrt{p\log(t/p)})$$

*Further, if $\|\Phi(x)\|_k \approx (\sqrt{m})$ grows with $m$, e.g. $\Phi(x) = 1_m$ (vector of ones) then*

$$\sqrt{2\log\left(\frac{1}{\delta}\frac{\det(\mathbf{\Omega}_{\lambda,t})^{1/2}}{\det(\lambda\mathbf{S})^{1/2}}\right)} = \mathcal{O}(\sqrt{p\log(tm/p\lambda + 1)})$$

*Proof.* First let us determine the absolute bound on all $\|z_i\|_2^2$. Due to projected data,

$$\mathbf{S}^{-1/2}z_i = (\mathbf{C}\mathbf{C}^\top)^{1/2}(\mathbf{C}\mathbf{C}^\top)^{-1}\mathbf{C}\Phi(x_i) \implies$$

$$\left\|\mathbf{S}^{-1/2}z_i\right\|_2^2 \le \left\|(\mathbf{C}\mathbf{C}^\top)^{1/2}(\mathbf{C}\mathbf{C}^\top)^{-1}\mathbf{C}\Phi(x_i)\right\|_2^2 \le \left\|\mathbf{C}^\top(\mathbf{C}\mathbf{C}^\top)^{-1}\mathbf{C}\right\|_2^2 L^2 = L^2.$$

where the last step follows from properties of projection. Notice also that $\frac{1}{\sigma^2}\operatorname{Tr}(\mathbf{S}^{-1/2}\mathbf{Z}^\top\mathbf{Z}\mathbf{S}^{-1/2}) = \frac{1}{\sigma^2}\sum_{i=1}^t \operatorname{Tr}(\mathbf{S}^{-1/2}z_i z_i^\top\mathbf{S}^{-1/2}) \le t\frac{L^2}{\sigma^2}$.

Using arithmetic-geometric mean inequality as in (Szepesvari and Lattimore, 2020) Lemma 19.4, we can show that

$$
\begin{aligned}
\frac{\det(\mathbf{\Omega}_{\lambda,t})}{\det(\lambda\mathbf{S})} &= \frac{\det(\mathbf{S}^{-1/2}\mathbf{\Omega}_{\lambda,t}\mathbf{S}^{-1/2})}{\det(\lambda\mathbf{I})} \\
&\le \frac{(\frac{1}{p}\operatorname{Tr}(\det(\mathbf{S}^{-1/2}\mathbf{\Omega}_{\lambda,t}\det(\mathbf{S}^{-1/2})))^p}{\lambda^p} = \frac{(\frac{1}{p}\operatorname{Tr}(\mathbf{S}^{-1/2}\mathbf{Z}\mathbf{Z}^\top\mathbf{S}^{-1/2} + \lambda\mathbf{I}))^p}{\lambda^p} \\
&\le \frac{(\frac{1}{p}(tL^2 + \lambda p))^p}{\lambda^p} \\
\log\left(\frac{\det(\mathbf{\Omega}_{\lambda,t})}{\det(\lambda\mathbf{S})}\right) &\le p\log\left(\frac{tL^2}{p\lambda} + 1\right)
\end{aligned}
$$

The last line of the Lemma follows from noting that $L^2 = \mathcal{O}(m)$.

$\square$

## B.4 Adaptive design and Modified Regularized Estimator: Relation to prior work

With the adaptive estimator, we could alternatively directly define an estimator for $\mathbf{C}\theta$ as $\mathbf{C}\hat{\vartheta}$ using only the projected values $\mathbf{Z}$ as done by Mutný et al. (2020). In this case, however, additional bias growing in time $t$ enters into the confidence parameter as (9) as $\frac{1}{\lambda}\sum_{i=1}^t \|j_i\|_{\mathcal{V}_0^{-1}}$, as we show in Theorem 10 below.

**Theorem 10** (Adaptive Design Regularized Regression - biased). *Let $\mathbf{Z}_t$ be s.t. $\mathbf{X}_i = \mathbf{Z}_i\mathbf{C} + \mathbf{J}_i$, where $\mathbf{J}_i$ is minimal as measured by squared norm, then the estimator*

$$\mathbf{C}\hat{\vartheta}_t = \arg\min_{\vartheta\in\mathbb{R}^k}\sum_{i=1}^n \frac{1}{\sigma^2}(y_i - \vartheta^\top z_i)^2 + \lambda\|\vartheta\|_{\mathbf{S}}^2 \tag{35}$$

*has anytime confidence sets for all $t \ge 0$*

$$\mathrm{P}\left(\left\|\mathbf{C}\theta - \mathbf{C}\hat{\vartheta}_t\right\|_{\mathbf{\Omega}_\lambda} \ge 1 + 2\log\left(\frac{1}{\delta}\frac{\det(\mathbf{\Omega}_{t,\lambda})}{\det(\tilde{\mathbf{V}}_0)}\right) + \sum_{i=1}^t \|\mathbf{J}_i\|_{\mathcal{V}_0^{-1}}\right) \le \delta \tag{36}$$

*where $\delta \in (0,1)$, and $\mathbf{\Omega}_{t,\lambda} = \frac{\mathbf{Z}_t^\top\mathbf{Z}_t}{\sigma^2} + \lambda\mathbf{S}$, where $\mathbf{S} = (\mathbf{C}\mathcal{V}_0\mathbf{C}^\top)^{-1}$*

Notice that if $\mathbf{J}$ is small in the Frobenius norm, the confidence sets improve. In fact, they depend only on the norm.

*Proof.*

$$\left\|\mathbf{C}\theta - \mathbf{C}\hat{\vartheta}\right\|_{\mathbf{\Omega}_\lambda}^2 = \left\|\mathbf{C}\theta - (\frac{1}{\sigma^2}\mathbf{Z}^\top\mathbf{Z} + \lambda\mathbf{S})^{-1}\mathbf{Z}^\top(\mathbf{X}\theta + \epsilon)\right\|_{\mathbf{\Omega}_\lambda}^2$$

$$
\begin{aligned}
&= \left\| \mathbf{C}\theta - \mathbf{\Omega}_\lambda^{-1}\mathbf{Z}^\top((\mathbf{ZC}+\mathbf{J})\theta + \epsilon) \right\|_{\mathbf{\Omega}_\lambda}^2 \\
&= \left\| \mathbf{C}\theta - \mathbf{\Omega}_\lambda^{-1}\frac{1}{\sigma^2}\mathbf{Z}^\top\mathbf{ZC}\theta + \frac{1}{\sigma^2}\mathbf{\Omega}_\lambda^{-1}\mathbf{Z}^\top\mathbf{J}\theta + \frac{1}{\sigma^2}\mathbf{\Omega}_\lambda^{-1}\mathbf{Z}^\top\epsilon \right\|_{\mathbf{\Omega}_\lambda}^2 \\
&= \left\| \mathbf{\Omega}_\lambda^{-1}(\mathbf{\Omega}_\lambda - \frac{1}{\sigma^2}\mathbf{Z}^\top\mathbf{Z})\mathbf{C}\theta + \frac{1}{\sigma^2}\mathbf{\Omega}_\lambda^{-1}\mathbf{Z}^\top\mathbf{J}\theta + \frac{1}{\sigma^2}\mathbf{\Omega}_\lambda^{-1}\mathbf{Z}^\top\epsilon \right\|_{\mathbf{\Omega}_\lambda}^2 \\
&= \left\| \mathbf{\Omega}_\lambda^{-1}\lambda\mathbf{SC}\theta + \frac{1}{\sigma^2}\mathbf{\Omega}_\lambda^{-1}\mathbf{Z}^\top\mathbf{J}\theta + \frac{1}{\sigma^2}\mathbf{\Omega}_\lambda^{-1}\mathbf{Z}^\top\epsilon \right\|_{\mathbf{\Omega}_\lambda}^2 \\
&\leq \underbrace{\left\| \mathbf{\Omega}_\lambda^{-1}\lambda\mathbf{SC}\theta \right\|_{\mathbf{\Omega}_\lambda}^2}_{\text{bias}} + \underbrace{\left\| \frac{1}{\sigma^2}\mathbf{\Omega}_\lambda^{-1}\mathbf{Z}^\top\mathbf{J}\theta \right\|_{\mathbf{\Omega}_\lambda}^2}_{\text{self-normalized bias}} + \underbrace{\left\| \frac{1}{\sigma^2}\mathbf{\Omega}_\lambda^{-1}\mathbf{Z}^\top\epsilon \right\|_{\mathbf{\Omega}_\lambda}^2}_{\text{self-normalized noise}}
\end{aligned}
$$

Let us now analyze each term separately. The self-normalized terms can be analyzed using a classical technique with a proper choice of mixture distribution as we show in the proof of the Theorem 3. In this case it is a normal distribution $\mathcal{N}(0, (\lambda\mathbf{S})^{-1})$.

The first term can be shown to be bounded by,

$$
\begin{aligned}
\left\| \lambda\mathbf{\Omega}_\lambda^{-1}\mathbf{SC}\theta \right\|_{\mathbf{\Omega}_\lambda}^2 &= \left\| \lambda\mathbf{\Omega}_\lambda^{-1}(\mathbf{S}(C\mathcal{V}_0\mathbf{C}^\top)\mathbf{S})\mathbf{C}\theta \right\|_{\mathbf{\Omega}_\lambda}^2 &&(37) \\
&= \lambda^2\theta^\top\mathbf{C}^\top\mathbf{S}\mathbf{\Omega}_\lambda^{-1}\mathbf{SC}\theta &&(38) \\
&\leq \lambda^2\theta^\top\mathbf{C}^\top\mathbf{S}(\lambda\mathbf{S})^{-1}\mathbf{SC}\theta &&(39) \\
&\leq \lambda\theta^\top\mathbf{C}^\top(\mathbf{C}\mathcal{V}_0\mathbf{C}^\top)^{-1}\mathbf{C}\theta &&(40) \\
&\leq \lambda\theta^\top\mathcal{V}_0\theta \leq 1 &&(41)
\end{aligned}
$$

They can be analyzed in the same spirit as in Mutný et al. (2020) novel term

$$
\begin{aligned}
\left\| \frac{1}{\sigma^2}\mathbf{\Omega}_\lambda^{-1}\mathbf{Z}^\top\mathbf{J}\theta \right\|_{\mathbf{\Omega}_\lambda}^2 &= \frac{1}{\sigma^2}\theta\mathbf{J}^\top\mathbf{Z}\mathbf{\Omega}_\lambda^{-1}\mathbf{Z}^\top\mathbf{J}\theta &&(42) \\
&= \theta\mathbf{J}^\top\frac{\mathbf{Z}}{\sigma}(\frac{\mathbf{Z}^\top\mathbf{Z}}{\sigma^2}+\tilde{\mathbf{V}}_0)^{-1}\frac{\mathbf{Z}^\top}{\sigma}\mathbf{J}\theta &&(43) \\
&\leq \theta\mathbf{J}^\top\mathbf{Z}(\mathbf{Z}^\top\mathbf{Z})^\dagger\mathbf{Z}^\top\mathbf{J}\theta &&(44) \\
&\leq \theta\mathbf{J}^\top\mathbf{J}\theta = \sum_{i=1}^t(j_i^\top\theta)^2 \leq \sum_{i=1}^t\|j_i\|_{\mathcal{V}_0^{-1}}^2\frac{1}{\lambda} &&(45)
\end{aligned}
$$

$\square$

# C Algorithms

Now we will briefly discuss algorithms that construct designs $\mathbf{X}$, and allocations over them $\eta$, which lead to low error either in expectation or with high probability - using $A$ or $E$ design, respectively. Depending on the aim and estimator, different algorithms might be preferable. For comprehensive reviews please refer to Todd (2016), Pukelsheim (2006),(Fedorov and Hackl, 1997) for optimization algorithms, and (Allen-Zhu et al., 2017), (Camilleri et al., 2021) for rounding techniques.

## C.1 Greedy selection

A first idea is to greedily maximize the scalarized information matrix with the update rule $\eta_{t+1} = \frac{t}{t+1}\eta_t + \frac{1}{1+t}\delta_t$,

$$
\delta_t = \arg\max_{x\in\mathcal{X}} f\left(\mathbf{W}_\lambda\left(\frac{t}{t+1}\eta_t + \frac{1}{t+1}\delta_x\right)\right)
$$

where $f$ refers to the scalarization and $\delta_x$ to the indicator of the discrete measure corresponding to the feature $\Phi(x)$. Notice that while stated in form of allocations, due to the form of the update rule $t\eta_t$ is always an integer.

Surprisingly, this algorithm can fail with the interpolation estimator, where adding a new row to $\mathbf{X}$, which does not lie in the kernel of $\mathbf{C}$ increases the variance. This is an artifact of the pseudo-inverse, but it demonstrates that the algorithm is not universal – hence we suggest using it with the regularized estimator only.

## C.2 Convex optimization

Alternatively, one can first select a design space $\mathbf{X}$ supported on finitely many queries such that if the optimal design is supported on it, it leads to a desired low bias. After that, we optimize the allocation $\eta \in \Delta^n$ over it. This has the advantage that *a)* we can bound the query complexity to reach $\epsilon$ of learning as in Proposition 14, and *b)* we can provably achieve optimality with convex optimization, in contrast to the greedy heuristic.

We look for an allocation using convex optimization methods as in (10) or more generally, $\max_{\eta \in \Delta^n} f(\mathbf{W}_\circ(\mathbf{D}(\eta)^{1/2}\mathbf{X}))$ where $\circ \in \{\dagger, \lambda\}$. There are three possible ways to initialize the algorithm with $\mathbf{X}$: make an ansatz, greedily reduce bias first, or use a modification of the random projection initialization of Betke and Henk (1992) described below.

**Mirror-descent and Regularity**   The objective above (or (10)) would be classically solved via Frank-Wolf algorithm (Todd, 2016), or a mirror descent algorithm (Beck and Teboulle, 2003) [2], which starts with the whole support, and reduces the weight of some of the queries $\eta_i$, as

$$\eta_{t+1} = \eta_t \frac{\exp(-s_t \nabla f(\mathbf{W}(\eta_t)))}{\sum_i \exp(-s_t \nabla_i f(\mathbf{W}(\eta_t)))}$$

where $s_t$ is the stepsize; if $s_t \propto \sqrt{t}^{-1}$, the convergence is guaranteed. Both $E$ and $A$-design objectives are concave as they are related by linear transform $\mathbf{C}$ from the classical objectives, which are known to be concave (Pukelsheim, 2006).

paragraphInitialization via random projections This algorithm is inspired by volume algorithm of Betke and Henk (1992). The algorithm proceeds by picking a random vector $c_0$ in the span of $\mathbf{C}^\top$ and picking two points $\bar{z}, \underline{z} = \arg\max_x c^\top x, \arg\min_x c^\top x$. Subsequently, we pick another $c_1$ which is orthogonal to $\underline{z} - \bar{z}$ and still in span of $\mathbf{C}^\top$. We repeat this procedure $k$ times. Should this not generate a sufficiently accurate starting point with desired bias (as measured by Def. 1), we repeat this procedure until such design is obtained.

## C.3 SDP reformulations of the objectives

We will now show that both $A$ and $E$ designs have an associated semi-definite reformulations which are possible if $\dim(\mathcal{H}_\kappa) = m$ [3]. One should bear in mind that these are possible only if the $\mathbf{C}\theta$ is estimable in the sense of Def. 3, Hence their might not be utilizable in many instances that this paper studies, and rather apply in cases classical experiment design. Nevertheless they do not appear in literature to the best of our knowledge.

On top of that these problems are more difficult to solve that lets say the alternative algorithm with mirror descent, however with the use of off-the shelf solvers for SDP problems such as cvxpy, they can be conveniently implemented. These refomrmulations are inspired by classical reformulations due to Boyd and Vandenberghe (2004), which show these for estimating $\theta$ directly. The key ingredient is to make use of generalized Shur-complement theorem of Zhang (2011).

**A-design**   objective is $\max_{\eta \in \Delta_n} 1/\operatorname{Tr}(\mathbf{W}(\eta)^{-1})$ can be represented as semi-definite program:

$$\min_{u \geq 0, \eta \in \Delta_n} \quad u_1 + u_2 + \cdots + u_k$$
$$\text{subject to} \quad \begin{pmatrix} \mathbf{V}(\eta) & \mathbf{C}^\top e_i \\ e_i^\top \mathbf{C} & u_i \end{pmatrix} \succeq 0 \ \text{ for all } i \in [k]$$

**E-design**   objective is $\max_{\eta \in \Delta_n} \lambda_{\min}(\mathbf{W}(\eta))$, can be again solved using semi-definite programming as:

$$\min_{t \in \mathbb{R}, \eta \in \Delta_n} \quad t$$

---

[2] Mirror descent is known as multiplicative algorithm in the experimental design literature (Silvey et al., 1978).

[3] We would like to thank Stephen Wright of University of Wisconsin for suggesting to revisit this idea.

$$\text{subject to} \qquad \begin{pmatrix} \mathbf{V}(\eta) & \mathbf{C}^\top \\ \mathbf{C} & t\mathbf{I}_{k \times k} \end{pmatrix} \succeq 0$$

We will show how to apply the generalized for the first objective (A-design) only. The second reformulation follows analogously. Notice that the objective $\max_\eta 1/\operatorname{Tr}(\mathbf{W}^{-1})$ is equivalent to $\min_\eta \operatorname{Tr}(\mathbf{W}^{-1})$, and $\mathbf{W}^{-1} = \mathbf{C}\mathbf{V}(\eta)^\dagger \mathbf{C}^\top$. Generalized Shur-complement lemma of Zhang (2011) states that $\mathbf{C}\mathbf{V}(\eta)^\dagger \mathbf{C}^\top \preceq \mathbf{\Lambda}$ with the projection operator lying in the null space of $\mathbf{C}$, i.e. $(\mathbf{I} - \mathbf{V}(\eta)^\dagger \mathbf{V}(\eta))\mathbf{C}^\top = \mathbf{0}$, is equivalent to

$$\begin{pmatrix} \mathbf{V}(\eta) & \mathbf{C}^\top \\ \mathbf{C} & \mathbf{\Lambda} \end{pmatrix} \succeq 0.$$

The second condition is satisfied as long as the problem is estimable, in other words, there exists $\mathbf{L}$ s.t. $\mathbf{C} = \mathbf{LX}$. Hence minimizing the trace of $\mathbf{\Lambda}$, we can minimize the trace of the information matrix. As the trace depends only on the diagonal elements of $\mathbf{\Lambda}$, we can choose it to be diagonal, i.e., $\mathbf{\Lambda} = \operatorname{Diag}(u)$.

### C.4 Query complexity and Optimization

The objective in (10) was stated in the kernelized form which is only valid if the RKHS is finite-dimensional. We state here for completeness the version which operates in the kernelized setting where all the operators can be inverted (and calculated explicitly using only the kernel $\kappa$ evaluation).

$$\eta^* = \underset{\eta \in \Delta^n}{\arg\max} \, \lambda_{\min} \left( \mathbf{C}\mathcal{V}_0^{-1}\mathbf{X}^\top (\mathbf{D}(\eta)\mathbf{X}\mathcal{V}_0^{-1}\mathbf{X}^\top\mathbf{D}(\eta))^{-2\dagger}\mathbf{X}\mathcal{V}_0^{-1}\mathbf{C}^\top \right)^{-1}. \tag{46}$$

## D Geometry

We saw in Section 6 that properties of $\{\Phi(x) : x \in \mathcal{S}\}$ enter into consideration when we want to bound the total number of queries required to reach $\epsilon$ accuracy with high probability such that we can learn $\mathbf{C}\theta$. The dependence enters as the smallest eigenvalue $\lambda_{\min}$ of the information matrix $\mathbf{W}$. In this section, we relate the eigenvalue $\lambda_{\min}(\mathbf{W})$ directly to the geometry of the above set using seminal results about E-experimental design from Pukelsheim and Studden (1993).

### D.1 Set Width

The geometrical property we are interested in *width of a set* that we define below. However, first, we want to make a general note that the formal results of our theorems hold for symmetrized sets $\{\Phi(x)|x \in \mathcal{S}\}$. This is without the loss of generality, since, an optimal design on symmetrized set and a non-symmetrized set is equivalent. This can be seen by considering $\mathbf{W}_1(\eta)$ defined on symmetrized set equates the one of non-symmetrized set $\mathbf{W}_2(\eta)$ due to the fact evaluations $\Phi(x)$ enter only as outer products $\mathbf{W}_1(\eta) = r(\sum_{i=1}^n \eta_i(\Phi(x_i)\Phi(x_i)^\top))$, where the sign clearly does not change anything in terms of the information matrix, and $r$ correspond to further operations to define $\mathbf{W}$ properly (i.e. projection and inverse).

**Definition 5** (Width). Let $\mathcal{D}$ be a convex set in $\mathcal{H}$, then the width of it is defined as,

$$\operatorname{width}(\mathcal{D}) = \min_{\theta \in \mathcal{H}} \max_{x,y \in \mathcal{D}} \theta^\top (x - y) \tag{47}$$

where $\theta$ is the unit norm.

*Width in the span of* $\mathbf{C}$ is defined to be,

$$\operatorname{width}(\mathcal{D}, \mathbf{C}) = \min_{\theta = \mathbf{C}^\top \alpha \text{ s.t. } \alpha \in \mathbb{R}^k} \max_{x,y \in \mathcal{D}} \theta^\top (x - y) \tag{48}$$

where $\alpha$ is the unit norm.

**Definition 6** (Gauge norm). Let $\mathcal{D}$ be a set of points in $\mathcal{H}$, then the gauge norm of $\mathcal{D}$ is,

$$\rho_{\mathcal{S}}(x) = \inf_{r \geq 0} \{r : x \in r \operatorname{conv}(\mathcal{D})\} \tag{49}$$

**Theorem 11** (Thm. 2.2 in (Pukelsheim and Studden, 1993)). *Let* $\mathbf{W}$ *be any the information matrix for* $\mathbf{C}\theta$*, and* $z \in \mathbb{R}^k$ *non-zero then*

$$\lambda_{\min}(\mathbf{W}) \leq \frac{\|z\|_2^2}{\rho(\mathbf{C}^\top z)^2}.$$

*It holds with equality if $\mathbf{W}$ has smallest eigenvalue of multiplicity one and $z$ is the associated eigenvector.*

The treatment without multiplicity is somewhat more complicated and can be found in.

**Proposition 12.** *Under assumption of Thm. 11, no multiplicty, $z$ being eigenvector of $\mathbf{W}$, let $\mathbf{C} : \mathbb{R}^m \to \mathbb{R}^p$, $\mathcal{D}$ be a symmetric convex set in $\mathcal{H}$, then*

$$\lambda_{\min}(\mathbf{W}) = \frac{\|z\|_2^2}{\rho(\mathbf{C}^\top z)^2} \geq \frac{1}{\mathrm{width}(\mathcal{S}, \mathbf{C})^2}$$

*Proof.* Due to homogenity of the gauge norm, $\rho(\mathbf{C}^\top z \frac{\|z\|_2}{\|z\|_2}) = \|z\|^2 (\rho \mathbf{C}^\top \hat{z})$, where $\hat{z}$ is a unit vector. Thus, the optimization can be restated as,

$$\frac{\|z\|_2^2}{\rho(\mathbf{C}^\top z)^2} = \frac{1.}{\rho(\mathbf{C}^\top \hat{z})^2} = \frac{1}{\mathrm{width}(\tilde{\mathcal{D}})^2} \geq \frac{1}{\mathrm{width}(\mathcal{D}, \mathbf{C}^\top \hat{z})^2}$$

where in the second to last step we use the fact that the set is symmetric and $\hat{z}$ is unit eigenvector. The last inequality follows by minimizing over all possible unit direction as int the definition of width. The set $\tilde{\mathcal{D}} \subset \mathcal{H}$ s.t. $\mathcal{D} = \{f \in \mathcal{H}_k | f = \mathbf{C}^\top \hat{z}\}$. Notice that this set lies in 1-dimension subspace, and is directly proportional to the gauge-norm. $\square$

Given the above theorem, we can calculate the worst-case value of parameter $\frac{1}{\lambda_{\min}(\mathbf{W})}$ that bounds the complexity if the geometry of the set can be understood easily. Alternatively, we can resort to direct calculation. For example consider the gradient design problem, in which we know that for finite difference problem can be bounded by $dh + O(h)$, where $h$ is the stepsize.

**Proposition 13.** *Under the assumption of the model* (1), *if $\mathbf{C} = \nabla_x \Phi(x)$ at $x \in \mathbb{R}^d$, $\Phi(x)$ is the evaluation functional, and the design space $\mathbf{X}$ contains rows $\{\Phi(x \pm he_i\}_{i=1}^d$ where $e_i$ corresponds to the unit vectors in $\mathbb{R}^d$ and $h$ is the step-size, then*

$$\frac{1}{\lambda_{\min}(\mathbf{W}(\eta))} \leq dh + O(h^2), \tag{50}$$

*where $\eta$ corresponds to the design supported on all points with equal weight.*

*Proof.* The inverse of the information matrix is,

$$
\begin{aligned}
\mathbf{W}_\dagger^{-1} &= \nabla_x \Phi(x) \left( \sum_{i=1}^d \eta_i (\Phi(x+he_i)\Phi(x+he_i)^\top + \Phi(x-he_i)\Phi(x-he_i)^\top) \right)^{-1} \nabla_x \Phi(x)^\top \\
&= \nabla_x \Phi(x) \left( \sum_{i=1}^d \eta_i (2\Phi(x)\Phi(x)^\top + h\nabla_x \Phi(x)^\top e_i e_i^\top \nabla_x \Phi(x) + O(h^2)) \right)^{-1} \nabla_x \Phi(x)^\top \\
&\preceq \nabla_x \Phi(x) \left( h\nabla_x \Phi(x)^\top \mathbf{D}(\eta) \nabla_x \Phi(x) + O(h^2) \right)^{-1} \nabla_x \Phi(x)^\top \\
&= h\nabla_x \Phi(x) \left( \nabla_x \Phi(x)^\top \mathbf{D}(\eta) \nabla_x \Phi(x) + O(h^2) \right)^{-1} \nabla_x \Phi(x)^\top \\
&= dh\nabla_x \Phi(x) \left( \nabla_x \Phi(x)^\top \nabla_x \Phi(x) + O(h^2) \right)^{-1} \nabla_x \Phi(x)^\top \\
&\preceq dh\mathbf{I} + O(h^2)
\end{aligned}
$$

In the first step we used Taylor's theorem and keep track of the order components. In the second we used that $\Phi(x)\Phi(x)^\top$ is psd, later we used the definition of the design and the properties of the projection matrix. Lastly,

$$\frac{1}{\lambda_{\min}(\mathbf{W}_\dagger)} = \lambda_{max}(\mathbf{W}_\dagger^{-1}) \leq dh + O(h^2),$$

which finishes the proof. $\square$

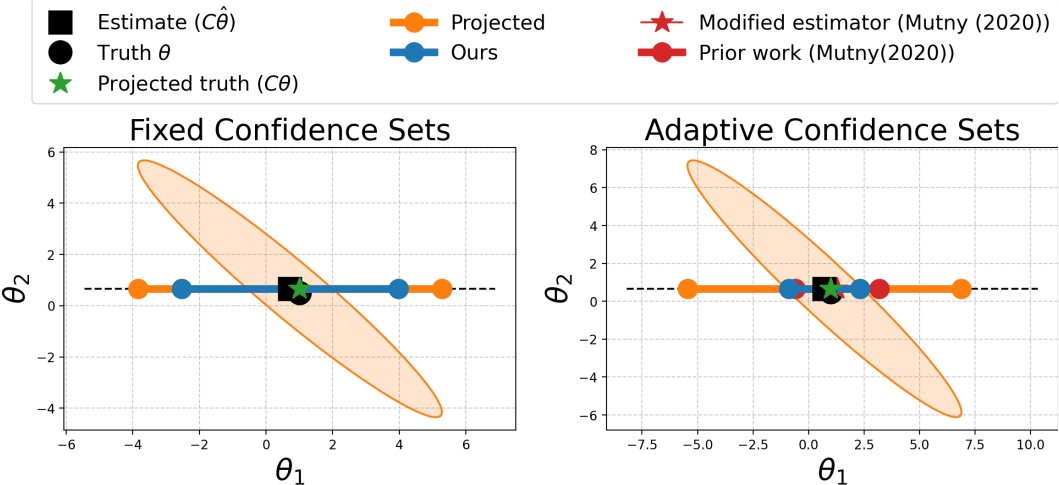

Figure 3: An example of confidence ellipsoids and intervals in two dimensions. You can see that in both regimes: fixed and adaptive. Our confidence sets (Ours) are the tightest for the estimation of the linear projection of $\mathbf{C}\theta$ value of the parameter $\theta = (1,1)$, in this case $\mathbf{C} = (1,0)$. On the left, we see fixed confidence sets while on the right are adaptive both with $T = 50$. We compare with (Mutný et al., 2020) and associated confidence sets, they are larger for this example (and grow with $T$); also the estimator is different as we delineate in Sec. B.4.

**Corollary 14** (Query complexity). *Under assumption of Prop. 1 where $\eta^*$ is the optimum to Eq. 10, it holds that*

$$\left\| \mathbf{C}(\hat{\theta}_\dagger - \theta) \right\|_2 \leq \sqrt{\frac{\lambda_{\min}(\mathbf{W}_\dagger(\eta^*))^{-1}}{T}} \left( \sigma\sqrt{\xi(\delta)} + \frac{\nu}{\sqrt{\lambda}} \right).$$

*with probability $1 - \delta$.*

*Proof.*

$$\left\| \mathbf{C}(\hat{\theta}_\dagger - \theta) \right\|_2^2 \quad \leq \quad \frac{1}{\lambda_{\min}(\mathbf{W}_\dagger(\eta^*))} \left\| \mathbf{C}(\hat{\theta}_\dagger - \theta) \right\|_{\mathbf{W}_\dagger(\eta^*)} \tag{51}$$

$$= \quad \frac{1}{\lambda_{\min}(\mathbf{W}_\dagger(\eta^*))} (\sigma^2 \xi(\delta) + \frac{\nu^2}{\sqrt{\lambda}}) \tag{52}$$

Now the $\mathbf{W}_\dagger$ nor $\nu^2$ do not depend on $\sigma^2$, however evaluating multiple times the same measurements Namely $T$ times, reduces the variance $\sigma^2$ by $1/T$. After taking this into consideration and the square root it finishes the proof. □

# E    Extra Examples

In this section, we provide additional applications of our framework that deserve a mention, but before we do, we want to bring attention to Figure 3, where we show a comparison of our confidence sets with prior work and projected ones, similar to Figure 1. As a refresher, recall that

$$\mathbf{L} : \mathbb{R}^n \to \mathbb{R}^p$$
$$\mathbf{X} : \mathcal{H}_k \to \mathbb{R}^n$$
$$\mathbf{C} : \mathcal{H}_k \to \mathbb{R}^p.$$

## E.1    Integral maps

**Quadrature**    An integral is a special linear operator on the function space,

$$\mathbf{C} = \int q(x)\Phi(x)^\top \cdot dx = \int q(x)\Phi(x)dx^\top\cdot,$$

where $p = 1$. The approximate estimability condition can be written in a concise form

$$\|\mathbf{C} - \mathbf{LX}\|_k = \left\| \int (q(x) - \sum_{y \in \mathcal{S}} l(y)\delta(x - y))\Phi(x)dx \right\|_k ,$$

where $l(y) = \mathbf{L}_y$, and $y \in \mathcal{S}$ which itself is equal to well-known integral distance $\mathrm{MMD}_{\mathcal{H}_\kappa}(q, \sum_{y \in \mathcal{D}} l(y)\delta(\cdot - y))$ referred to as Maximum Mean Discrepancy (MMD) multiplied by $B$. It measures the worst case integration error on the class of functions $\theta \in \mathcal{H}_\kappa$ s.t. $\|\theta\|_k \leq 1$ with nodes $x \in \mathcal{S}$ and weights $l(x)$. The method which minimizes this quantity is sometimes referred to as Bayesian Quadrature (Huszár and Duvenaud, 2012). Note that minimization of this quantity would correspond to the interpolation estimator.

**Fourier Spectra: Low-pass Filters**  Suppose we are interested in approximately learning the Fourier spectrum of $\theta$. This might be especially interesting if $\theta$ is composed of two signals one with specific low-frequency bands and the rest occurs only elsewhere and is of no interest. Let $\{\omega_i\}_{i=1}^p$ be the frequencies of interest, then the rows of $\mathbf{C}$ are $\mathbf{C}_j = \int_\infty^\infty \exp(i\omega_j x)\Phi(x)^\top \cdot dx$, which can be discretized or evaluated in closed form depending on the kernelized space.

**Transductive Risk**  If the dimension of the response vector is very large much larger than the number of elements in the unlabeled set, which is a common appearance in deep learning or high dimensional statistics; there is still hope to have provable bounded error on selected important candidates. As oposed to study the full risk $\mathbb{E}_\epsilon[\mathbb{E}_{x \sim \rho}[(x^\top(\theta - \hat\theta))^2]]$ with data distribution $\rho$, we can optimize the risk on a set of candidates $\mathbb{E}_\epsilon[\sum_i^p[(q_i^\top(\theta - \hat\theta))^2]]$, which becomes $\mathrm{Tr}(\mathbf{Q}\mathbf{V}^\dagger\mathbf{Q})$, where $\mathbf{Q}$ contains the rows of $k$ are the candidates.

## E.2  Solution to Contamination (Low pass filter)

Often in estimation, the signal of interested is corrupted by another signal that we are not interested in. This is sometimes referred to as *contamination* (Fedorov and Hackl, 1997). In particular, consider the following model $\mathbb{E}[y] = (\alpha \quad \beta)^\top (\phi(x) \quad \psi(x))$, where we want to infer $\alpha$ only. Note that this is a special case of our framework, where the linear functional $\mathbf{C} = \mathbf{I}_S$ zeros out the variables $\beta$ in $\theta = (\alpha, \beta)$. This problem is well-defined only if the spans of $\{\phi(x)\}_{x \in \mathcal{S}}$ and $\{\psi(x)\}_{x \in \mathcal{S}}$ are not contained in each other, which we neglect for our purposes here.

For illustration, consider a linear trend contaminated with function $f$, which has major frequency components that are *known*, i.e., $y = \alpha^\top x + f(x) + \epsilon$. We can stack these frequencies in an evaluation functional $\psi(x, \{\omega\}_{i=1}^m)$, and express the problem in the form above. In Fig. 5, we report the MSE for estimation of $\alpha$ with $f$ which has frequency components $\{\pi l, \pi el | l \in [16]\}$ weighted with $1/l^2$. The contamination-aware design tries to query points that eliminate the effect of $f$ and leads to lower MSE for the same number of observations and performs much better in expectation than either random or full designs – an optimal design that learns both $\alpha$ and $\beta$. For this design, we chose to sample greedily first to reduce bias and then optimized the weights of each data point to achieve optimality.

## E.3  Learning ODE solutions

Consider an example of damped harmonic oscillator, $u(t)'' + \kappa u(t)' + u(t) = 0$ for $t \in [t_0, t_1]$. Adopting the above procedure and using the squared exponential kernel to embed trajectories [4] , we show in Figure 4a inference with the shape constraint using $A$-optimal design as well as equally space design, we see the resultant confidence bands and fit are much better for optimized design. We assume that $\kappa \in (\kappa_-, \kappa_+)$ and give a union of confidence sets as in the robust design case. The optimal design then corresponds to the unknown null subspace $\mathbf{C}$ and its variance part is equal to $\sup_{\kappa \in [\kappa_-, \kappa_+]} \mathrm{Tr}(\mathbf{C}_\kappa \mathbf{V}^\dagger \mathbf{C}_\kappa^\top)$.

## E.4  Learning PDE solutions

In a similar fashion as the linear ODE feature, a linear constraint so do linear PDEs. We demonstrate this on a sensor placement problem, where consider heat equation in two dimensions with $u(t, x)$ with varying diffusion coefficient $c(x)$.

$$\partial_t u - c(x)\partial_{xx} u = 0 \quad \text{in} \quad (0, T) \times \Omega$$

---

[4]The nullspace of $\mathbf{Z}$ forms its own Hilbert space that spans the space of trajectories, however for estimation convenience it is sometimes easier to work with a larger space and then project.

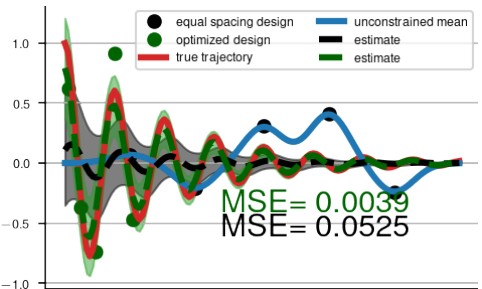
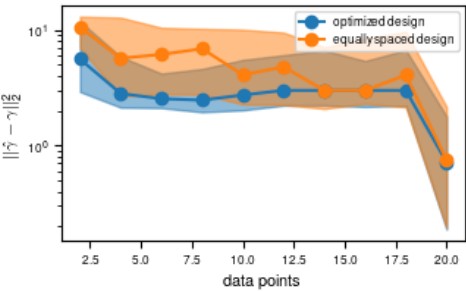

(a) An estimate of the trajectory of a damped harmonic oscillator. The source of unknown here is due to the unknown initial conditions. Notice that with the optimized design we can achieve a much better fit. Also, the unconstrained fit with the squared exponential kernel is shown in blue. Below we also see the reduction in MSE for estimating this trajectory with the two designs.

(b) MLE error of estimating $\gamma$ in the pharmacokinetic study. We see that with the optimized design we are consistently better in terms of MSE than equally spaced design. Error bars with 20 repetitions included. The y-axis denotes the number of data samples (draws of blood) required. Notice the logarithmic scale.

Figure 4: Further experiments showing application in ODE trajectory and parameter estimation.

$$u(0, x) = \langle \theta, \Phi(x) \rangle \quad \text{for} \quad x \in \Omega, z \in \partial\Omega.$$

We assume that the point value of initial conditions $u(0, x)$ can be measured by placing sensors along $x$. The goal is to place these sensors such that after elapsed time $T$, $u(T)$ can be inferred with the lowest error. This is the forward formulation of the inverse problem in Leykekhman et al. (2020). We assume that $\theta$ is a bounded member of RKHS due to the squared exponential kernel. In this case we are not interested in the full nullspace of $\mathbf{N}$ as in the previous example, instead we focus on the range space of $\mathbf{C} = \Phi(x, T)(\mathbf{N}^\top \mathbf{N})$ corresponding to information at $\Phi(x, T)$.

### E.5 Sequential Design: Estimating CVar

Often the objective we are trying to estimate is inherently stochastic such as the following model:

$$\mathbb{E}_{z \sim W(x)}[\rho(\Phi(x, z)^\top \theta)]. \tag{53}$$

In particular one can be interested in $x$ s.t. $\arg\sup_x \mathbb{E}_{z \sim W}[\Phi(x, z)^\top \theta]$. This is analyzed in sequential experiment design optimization with risk measures, where $\rho$ is the risk measure (see (Cakmak et al., 2020) and (Agrell and Dahl, 2020)). It is often assumed that $W$ cannot be evaluated explicitly, only sampled from with the evaluation oracle $y_i = \Phi(x_i, z_i)^\top + \epsilon_i$. We focus on a problem where we want to learn the risk of the actions $q \in Q, |Q| = k, \mathbb{E}_{z \sim W}[\rho(\Phi(q, z)^\top \theta)]$ instead of minimization and $\rho$ is CVar risk measure. With a fixed value $\theta$, this is a linear operator $\mathbf{C}_\theta$. Since $\theta$ is unknown, we adopt a sequential procedure, where we solve for the risk $C_{\hat{\theta}}$, where $\hat{\theta}$ is sampled uniformly from its confidence set.

## F Experiments: Details

### F.1 Gradient Estimation: Details

With gradient estimation we are interested in $\nabla_x \Phi(x)$ of an element $\theta \in \mathcal{H}_\kappa$. The example generated in Fig. 2b is in 2 dimensions, where we approximate $\Phi(x)$ of squared exponential kernel with lengthscale $l = 0.1$ with Quadrature Fourier Features of Mutný and Krause (2018) for the convenience of optimization. We consider a slightly off-set finite difference design, with the following elements,

$$\{\Phi(x), \Phi(x + he_1), \Phi(x + 2he_1), \Phi(x - he_1), \Phi(x - he_2)\}.$$

We calculate the relative-bias $\nu$ due to the Def. 1 and balance it with the budget (as in the Figure). This way we can identify the optimal $h^*$ before even solving for the optimal design. Given the identified $h^*$, we calculate the optimal design with mirror descent solving the objective (10), to get

$$\eta^* = (0.37, 0.09, 0.08, 0.09, 0.38),$$

which is not obvious without understanding the geometry of the problem. The value of $\sigma$ was set to be 0.01. The total error in Fig. 2b is the combination of bias and variance which was calculated using the largest eigenvalue of $\mathbf{E}(\mathbf{L}_\dagger)$ as in (2). This is the error we can guarantee with high probability (not the actual error) which is somewhat lower.

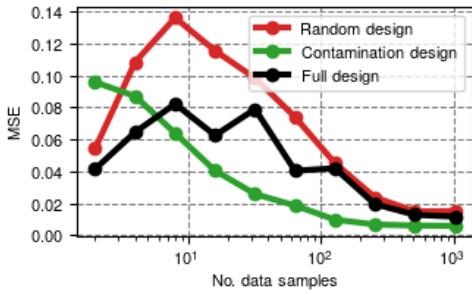 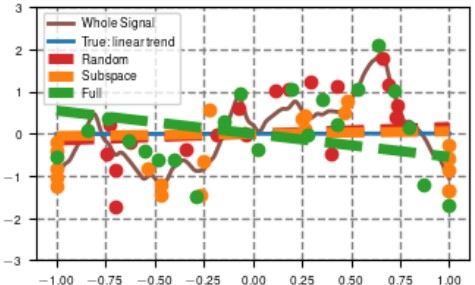

(a) Mean squared error of the estimated linear trend. We see that contamination-aware design improves over full design as well as randomly selected design.

(b) Example of the signal in brown with design points as well as linear trend estimates in color-coding. Full denotes joint estimation of brown and blue signal. Random designates a random design.

Figure 5: Statistical contamination: a numerical experiment

## F.2 The solution to Contamination (Statistics): Details

For this example, consider a linear trend contaminated with $f(x)$ which has known major frequency components, i.e., $y = \alpha^\top x + f(x) + \epsilon$. We can concatenate these frequencies, and construct an evaluation operator where $\Psi(x)_i = \cos(\omega_i x)$, and $\Psi(x)_{i+1} = \sin(\omega_i x)$ where $\omega_i$ is one of these frequencies. Then, we express the problem as $y = \theta^\top \Phi(x) = (\alpha, \beta)^\top (x, \Psi(x, \{\omega_i\}_{i=1}^m) + \epsilon$. Since we are interested only in $\alpha$, the operator $\mathbf{C} = (1, \mathbf{0})$ selects it. Hence we look for a design (subset of points) which only minimizes the residuals due to estimation of $\alpha$. In fact, such design might indirectly minimize the other residuals as well, but only if this is necessary to reduce the residuals due to $\alpha$.

The example depicted in Fig. 5 is created using the frequencies $\{\pi l | l \in [16]\} \cup \{\pi el | l \in [16]$ weighted with $1/l^2$. Such weighting of high frequencies creates contaminating signals that are similar to functions from Matérn kernel spaces (Rasmussen and Williams, 2006). The contamination-aware design performs much better.

The values of $\sigma = 0.5$ (relatively high). We compare with full design – one that minimizes the overall estimation error or in other words, where $\mathbf{C}$ is the identity. We first use the greedy algorithm to minimize the regularized estimator to obtain initial design space. Then we use convex optimization to find a proper allocation of a fixed budget (varying in the x-axis) to these queries. We again optimize with mirror descent.

## F.3 Pharmacokinetics: Details

The pharmacokinetic two-compartment model models two organs. In this case the stomach and blood and the transfer of the medication between them. The model can be mathematically described as

$$\frac{d}{dt}c_s(t) = -ac_s(t)$$
$$\frac{d}{dt}c_b(t) = bc_s(t) - dc_b(t)$$

where $c_s(t)$ represent concentration of the medication in the stomach, and $c_b(t)$ represents the concentration in the blood. The goal of pharmacokinetic studies is to identify $(a, b, c)$ from draws of the blood of subjects to e.g. properly assign dosage.

In comparison to the example of the damped harmonic oscillator, the initial conditions are known – albeit imprecisely – what is the true unknown is $\gamma = (a, b, c)$. We make the following consideration, in order to infer $\gamma$ precisely we need to have a precise estimation of the trajectory, hence we inject a slight perturbation to the initial condition and then search for the optimal design that is good for any of the models $\gamma$: robust design. This way with any of the models from $\gamma \in \Gamma$ we will have a good estimation of trajectories and hence a good estimation of $\gamma$ using MLE as we report in Fig. 4b.

The specific parameter used was $\sigma = 0.01$ which corrupted the observations of $c_b(t)$ at chosen times and the space which Embeds the trajectories was chosen to be that of the squared exponential kernel with lengthscale $l = 0.05$. The true parameters were set to be $(5, 10, 10)$ and the robust set

of $\Gamma = ((4, 6), (9, 11), (9, 11))$ for each parameter. We use the greedy algorithm with regularized estimator and $\lambda = 1/2$ to optimize the robust $A$-optimal metric. The resultant set is visualized in Fig. 2a.

### F.4 Control: Details

In the last experiment, we essentially model the same problem as in Lederer et al. (2021) as we explain. The only difference in our setup is the use of a gain $K = 200$ instead of $K = 15$. With a low gain of $K = 15$ one cannot provably stabilize with the given Lyapunov function. With $K = 200$ we were able to quickly stabilize the system once enough data points were obtained. Notice the reference trajectory and trajectory with "bad" and "good" controllers in Fig. 6. The driving non-linear dynamics are visualized likewise.

The experiment is designed to showcase the merit of the newly designed adaptive confidence sets with the stopping rule that stops once the controller has been verified to be stable. We repeated each experiment 10 times and reported the standard quantiles of the total derivative of the Lyapunov function in Fig. 2c. We always started with the initial set of 10 data points and then proceeded with exploration (adding specific data points) if the supremum of the total derivative was not negative. When it was we stopped. The stopping corresponds to the line going into negative values in the log plot in Fig. 2c. We considered these four strategies:

- *random* - randomly sampling data point from the whole domain [-1.5,1.5].

- *random-ref* - randomly sampling data point around the operating region.

- *unc* - sampling (greedily) the most uncertain query from the whole domain [-1.5,1.5].

- *unc-ref* - sampling (greedily) the most uncertain query from the operating region.

Berkenkamp et al. (2016) provides a better solution than uncertainty sampling, which is a special case of the linear functional studies in this work. In this example, this closely follows the *unc-ref* baseline and would create additional clutter. The inclusion of this algorithm would not further validate the benefit of the tighter confidence sets. The operating region is chosen to be a tube around the reference trajectory of width $h = 0.01$.

The dynamics is modeled using Nystöm features ($m = 400$) of the squared exponential kernel with lengthscale 0.25 and regularized estimator with $\lambda$ estimated experimentally. We selected this with visual inspection since the example is known, but one can adopt the procedure in Umlauft et al. (2018). The noise std. $\sigma = 0.05$ and number of queries is $T = 500$.

## G  Improved regret of linear bandits: Proofs

To prove the following theorem we require the famed potential lemma which we state for completeness,

$$\sum_{t=1}^{T} \|x_t\|_{\mathbf{V}_t^{-1}} \le \mathcal{O}(\sqrt{dT \log(T/d)}).$$

A reference for this result can be found in (Szepesvari and Lattimore, 2020). Note that a bounded pay-off vector assumption has been used in the derivation. In general this result can be extended to kernelized bandits by noting that

$$\sum_{t=1}^{T} \|x_t\|_{\mathbf{V}_t^{-1}} = \mathcal{O}(\sqrt{\gamma_T T}),$$

where $\gamma_T = \log\left(\frac{\det(\mathbf{V}_t)}{\det(\mathbf{I})}\right)$, due to matrix inversion lemma, this can be also expressed in kernelized form, as is usually referred to as information gain (Srinivas et al., 2010).

**Theorem 15.** *Let $\theta \in \mathbb{R}^d$ be an unknown pay-off vector, and a set of actions $x \in \mathcal{X}$ such that $|\mathcal{X}|\infty$, then the regret of UCB algorithm is no more than*

$$R_T \le \mathcal{O}(\sqrt{dT \log(T(|\mathcal{X}| + 1)/\delta)}) \tag{54}$$

*with probability $1 - \delta$.*

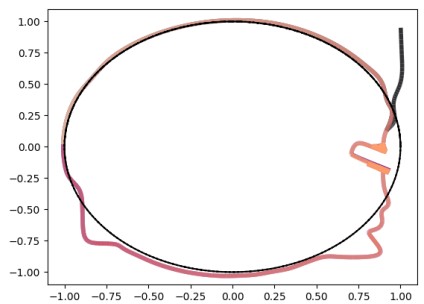 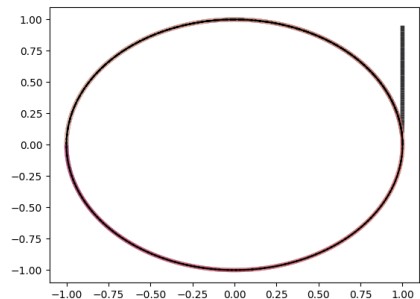

(a) An example of an unstable trajectory around the reference trajectory, started from (1,1) point

(b) An example of a stable trajectory around the reference trajectory, started from (1,1) point

 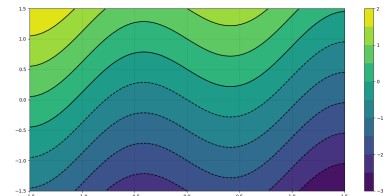

(c) Non-linear dynamics function $f_1(x)$ which governs the first component $x$.

(d) Non-linear dynamics function $f_2(x)$ which governs the second component of $x$.

Figure 6: Control Experiment: Additional details.

*Proof of Theorem 15.* We drop $\Phi$ and use $x$ instead only. Note that due to Lemma 3, we have that

$$\left\| x_k^\top (\hat{\theta} - \theta) \right\|_{\mathbf{W}_{t,k}}^2 \leq \left\| x_k^\top (\hat{\theta} - \theta) \right\|_{\mathbf{\Omega}_{t,k}}^2 \leq \beta_t,$$

where $\beta_t$ is defined for the information matrix $\mathbf{\Omega}_{t,k}$ in Theorem 3. The subscript $k$ designates the dependence on $x_k$ as we will need to consider all $x_k \in \mathcal{X}$, where we index $\mathcal{X}$ using $k \in \mathcal{N}$. Using this and $\delta_{k,t}$ indicator which is zero or one depending on whether action $k$ was played at time $t$. Note that,

$$
\begin{aligned}
R_T \quad &\leq \quad \sum_{t=1}^{T} (x)^\top \theta - x^\top \theta = \sum_{t=1}^{T} (x)^\top \theta - x_t^\top \tilde{\theta}_t + x_t^\top \tilde{\theta}_t - x_t^\top \theta \\[2mm]
&\overset{\text{def. UCB}}{\leq} \quad \sum_{t=1}^{T} x_t^\top (\tilde{\theta}_t - \theta) = \sum_{t=1}^{T} \sum_{k=1}^{K} \delta_{k,t} x_t^\top (\tilde{\theta}_t - \theta) \leq \sum_{t=1}^{T} \sum_{k=1}^{K} \delta_{k,t} |x_t^\top (\tilde{\theta}_t - \theta)| \\[2mm]
&= \quad \sum_{t=1}^{T} \sum_{k=1}^{K} \delta_{k,t} |x_t^\top (\tilde{\theta}_t - \theta) \mathbf{W}_{t,k}^{1/2} \mathbf{W}_{t,k}^{-1/2}| = \sum_{t=1}^{T} \sum_{k=1}^{K} \delta_{k,t} |x_t^\top (\tilde{\theta}_t - \theta) \mathbf{W}_{t,k}^{1/2}| |\mathbf{W}_{t,k}^{-1/2}| \\[2mm]
&= \quad \sum_{t=1}^{T} \sum_{k=1}^{K} \delta_{k,t} \underbrace{\sqrt{x_k^\top \mathbf{V}_t^{-1} x_k}}_{\mathbf{W}_{t,k}^{-1/2}} \left\| x_k^\top (\tilde{\theta}_t - \theta) \right\|_{\mathbf{W}_{t,k}} \leq \sum_{t=1}^{T} \sum_{k=1}^{K} \delta_{k,t} \sqrt{x_k^\top \mathbf{V}_t^{-1} x_k} \left\| x_k^\top (\tilde{\theta}_t - \theta) \right\|_{\mathbf{\Omega}_{t,k}} \\[2mm]
&\overset{\text{Thm. 3}}{\leq} \quad \sum_{t=1}^{T} \sum_{k=1}^{K} \delta_{k,t} \sqrt{x_k^\top \mathbf{V}_t^{-1} x_k} \beta_{t,k} \\[2mm]
&\overset{\text{Lemma 4}}{\leq} \quad \sum_{t=1}^{T} \sum_{k=1}^{K} \delta_{k,t} \| x_k \|_{\mathbf{V}_t^{-1}} \sqrt{\log(|\mathcal{X}| T/\delta)} \\[2mm]
&\overset{\text{Potential Lemma}}{\leq} \quad \sqrt{\log(|\mathcal{X}| T/\delta)} \sum_{t=1}^{T} \| x_t \|_{\mathbf{V}_t^{-1}} \leq \mathcal{O}\left( \sqrt{dT \log((|\mathcal{X}| + 1) T/\delta)} \right)
\end{aligned}
$$

which finishes the proof. The $\tilde{\theta}_t$ is the vector that corresponds to the UCB action played by the algorithm. We took the union bound for the adaptive confidence bounds for each $x \in \mathcal{X}$, and hence the logarithmic dependence on $|\mathcal{X}|$. $\qquad\square$

# H  Matrix Algebra Results

**Definition 7** (Matrix slices and weighted slices)**.**

$$\mathbf{M}_{SS} \stackrel{\text{def}}{=} \mathbf{I}_{:S}^\top \mathbf{M} \mathbf{I}_{:S} \quad \text{and} \quad \mathbf{M}_{S\eta>0} = \mathbf{D}(\eta > 0)\mathbf{M}\mathbf{D}(\eta > 0) \tag{55}$$

$$\mathbf{M}_S \stackrel{\text{def}}{=} \mathbf{I}_{:S}\mathbf{M}_{SS}\mathbf{I}_{:S}^\top \quad \text{and} \quad \mathbf{M}_{S\eta} = \mathbf{D}(\eta)\mathbf{M}\mathbf{D}(\eta) \tag{56}$$

**Lemma 5** (Zhang (2011))**.** *Let M be a positive definite matrix, and S be a subset of $[n]$, then*

$$(\mathbf{M}_S)^\dagger \preceq (\mathbf{M}^{-1})_S \tag{57}$$

**Lemma 6** (Zhang (2011))**.** *If $\mathbf{A} \succeq \mathbf{B}$ where $\mathbf{A}, \mathbf{B} \in \mathbb{R}^{l \times l}$ then for any $X \in \mathbb{R}^{d \times l}$, $\mathbf{X}\mathbf{A}\mathbf{X}^\top \succeq \mathbf{X}\mathbf{B}\mathbf{X}^\top$.*

**Lemma 7.** *Let $\mathbf{A} \in \mathbb{R}^{k \times m}$, then the matrix $\mathbf{P} = \mathbf{A}^\top (\mathbf{A}\mathbf{A}^\top)^{-1}\mathbf{A}$ is projection matrix.*

*Proof.* Its easy to check $\mathbf{P}^\top = \mathbf{P}$ and $\mathbf{P}^2 = \mathbf{P}$. $\qquad\square$

**Lemma 8.** *Let $\mathbf{A} \in \mathbb{R}^{d \times d}$ full rank, and $\mathbf{C} \in \mathbb{R}^{k \times d}$ with rank $k$. Then $\mathbf{C}\mathbf{A}\mathbf{C}^\top$ has rank $k$.*

**Proposition 16.** *Let $\mathbf{X} \in \mathbb{R}^{m \times n}$ s.t. $\min(m, n) \geq |\eta|$, where $\mathbf{X}$ is full rank,*

$$\mathbf{V}(\eta)^\dagger = \mathbf{X}^\top \mathbf{D}(\eta)((\mathbf{X}\mathbf{X}^\top)_{S\eta})^\dagger \mathbf{D}(\eta)((\mathbf{X}\mathbf{X}^\top)_{S\eta})^\dagger \mathbf{D}(\eta)\mathbf{X}$$

*Also,*

$$\mathrm{Tr}(\mathbf{V}^\dagger) = \mathrm{Tr}(\mathbf{D}(\eta)^{1/2}((\mathbf{X}\mathbf{X}^\top)_{S\eta})^\dagger \mathbf{D}(\eta)^{1/2})$$

*If $n = |\eta|$, then*

$$\mathbf{V}(\eta)^\dagger = \mathbf{X}^\top (\mathbf{X}\mathbf{X}^\top)^{-1}\mathbf{D}(\eta)^{-1}(\mathbf{X}\mathbf{X}^\top)^{-1}\mathbf{X} = \mathbf{X}^\dagger \mathbf{D}(\eta)^{-1}(\mathbf{X}^\top)^\dagger. \tag{58}$$

*Proof.* Tedious verification of four pseudo-inverse criteria. $\qquad\square$

## H.1  Auxiliary

**Lemma 9.** *Let $\mathbf{A} \in \mathbb{R}^{l \times l} \succ 0$, and $\mathbf{X} \in \mathbb{R}^{d \times l}$ diagonal where $d \leq l$, then*

$$\mathbf{A} \succeq \mathbf{X}^\top (\mathbf{X}\mathbf{A}^{-1}\mathbf{X}^\top)^{-1}\mathbf{X}. \tag{59}$$

*Proof.* Let $S$ be a set that selects exactly the non-zero elements of matrix $\mathbf{X}$ and further suppose w.l.o.g. the matrix is in such permutations that this block is right-upper most.

By Lemma 5, we know that $(\mathbf{A}_{SS})^{-1} \preceq (\mathbf{A}^{-1})_{SS}$. Due to the monotonicity of the operation (Lemma 6) $\mathbf{X}_{SS}(\mathbf{A}_{SS})^{-1}\mathbf{X}_{SS}^\top \preceq \mathbf{X}_{SS}(\mathbf{A}^{-1})_{SS}\mathbf{X}_{SS}^\top$. Inverse operator reverse the above inequality, hence,

$$\begin{align}
(\mathbf{X}_{SS}(\mathbf{A}_{SS})^{-1}\mathbf{X}_{SS}^\top)^{-1} &\succeq (\mathbf{X}_{SS}(\mathbf{A}^{-1})_{SS}\mathbf{X}_{SS}^\top)^{-1} \tag{60} \\
(\mathbf{X}_{SS}^{-\top}(\mathbf{A}_{SS})\mathbf{X}_{SS}^{-1}) &\succeq (\mathbf{X}_{SS}(\mathbf{A}^{-1})_{SS}\mathbf{X}_{SS}^\top)^{-1} \tag{61} \\
(\mathbf{Y}_{SS}^\top(\mathbf{A}_{SS})\mathbf{Y}_{SS}) &\succeq (\mathbf{X}_{SS}(\mathbf{A}^{-1})_{SS}\mathbf{X}_{SS}^\top)^{-1} \tag{62} \\
(\mathbf{A}_{SS}) &\succeq \mathbf{X}_{SS}^\top(\mathbf{X}_{SS}(\mathbf{A}^{-1})_{SS}\mathbf{X}_{SS}^\top)^{-1}\mathbf{X}_{SS} \tag{63}
\end{align}$$

where definite following matrix $\mathbf{Y}$ s.t. $\mathbf{Y}_{SS} = (\mathbf{X}_{SS})^{-1}$ and zero-otherwise.

Lastly, we know that $\mathbf{A}_S$ has eigenvalues strictly smaller than $\mathbf{A}$, and also note that $\mathbf{X}\mathbf{A}^{-1}\mathbf{X}^\top = \mathbf{X}_{SS}(\mathbf{A}^{-1})\mathbf{X}_{SS}^\top$. Thus applying Lemma 6 we can lift the expression to $l \times l$ matrices,

$$(\mathbf{A}) \succeq \mathbf{I}_{:S}(\mathbf{A})_{SS}\mathbf{I}_{S:} \succeq \mathbf{I}_{:S}\mathbf{X}_{SS}^\top(\mathbf{X}_{SS}(\mathbf{A}^{-1})_{SS}\mathbf{X}_{SS}^\top)^{-1}\mathbf{X}_{SS}\mathbf{I}_{S:} \succeq \mathbf{X}^\top(\mathbf{X}_{SS}(\mathbf{A}^{-1})_{SS}\mathbf{X}_{SS}^\top)^{-1}\mathbf{X}$$

$$= \mathbf{X}(\mathbf{X}\mathbf{A}^{-1}\mathbf{X})^{-1}\mathbf{X}$$

□

**Theorem 17.** *Let* $\mathbf{A} \in \mathbb{R}^{l \times l} \succ 0$*, and* $\mathbf{M} \in \mathbb{R}^{d \times l}$*, where* $d \leq l$*, and* $\mathrm{rank}(\mathbf{M}) = d$ *then*

$$\mathbf{M}^\top(\mathbf{M}\mathbf{A}^{-1}\mathbf{M}^\top)^{-1}\mathbf{M} \preceq \mathbf{A} \tag{64}$$

*Proof.* Let $\mathbf{M} = \mathbf{U}\mathbf{S}\mathbf{V}^\top$, where $\mathbf{U} \in \mathbb{R}^{d \times d}$ and $\mathbf{V} \in \mathbb{R}^{l \times l}$, be an SVD decomposition of $\mathbf{M}$ and $\mathbf{A} = \mathbf{R}^\top\mathbf{\Lambda}\mathbf{R}$, where $\mathbf{R} \in \mathbb{R}^{l \times l}$. be eigendecomposition of $\mathbf{A}$.

We perform the set of following equivalent operations to reduce the problem to diagonal case,

$$\begin{align}
\mathbf{R}^\top\mathbf{\Lambda}\mathbf{R} &\succeq (\mathbf{U}\mathbf{S}\mathbf{V}^\top)^\top(\mathbf{U}\mathbf{S}\mathbf{V}^\top\mathbf{R}^\top\mathbf{\Lambda}^{-1}\mathbf{R}(\mathbf{U}\mathbf{S}\mathbf{V}^\top)^\top)^{-1}\mathbf{U}\mathbf{S}\mathbf{V}^\top \tag{65} \\
\mathbf{R}^\top\mathbf{\Lambda}\mathbf{R} &\succeq \mathbf{V}\mathbf{S}^\top\mathbf{U}^\top(\mathbf{U}\mathbf{S}\mathbf{V}^\top\mathbf{R}^\top\mathbf{\Lambda}^{-1}\mathbf{R}\mathbf{V}\mathbf{S}^\top\mathbf{U})^{-1}\mathbf{U}\mathbf{S}\mathbf{V}^\top \tag{66} \\
\mathbf{R}^\top\mathbf{\Lambda}\mathbf{R} &\succeq \mathbf{V}\mathbf{S}^\top(\mathbf{S}\mathbf{V}^\top\mathbf{R}^\top\mathbf{\Lambda}^{-1}\mathbf{R}\mathbf{V}\mathbf{S}^\top)^{-1}\mathbf{S}\mathbf{V}^\top \tag{67} \\
\mathbf{\Lambda} &\succeq \mathbf{G}\mathbf{S}^\top(\mathbf{S}\mathbf{G}^\top\mathbf{\Lambda}^{-1}\mathbf{G}\mathbf{S}^\top)^{-1}\mathbf{S}\mathbf{G}^\top \tag{68} \\
\mathbf{B} &\succeq \mathbf{S}^\top(\mathbf{S}\mathbf{B}^{-1}\mathbf{S}^\top)^{-1}\mathbf{S} \tag{69}
\end{align}$$

where $\mathbf{V} = \mathbf{R}^\top\mathbf{G}$ is a new rotation matrix, and $\mathbf{B} = \mathbf{G}^\top\mathbf{\Lambda}\mathbf{G} = \mathbf{V}^\top\mathbf{R}^\top\mathbf{\Lambda}\mathbf{R}\mathbf{V} = \mathbf{V}^\top\mathbf{A}\mathbf{V}$. The rest follows from Lemma 9. □

**Proposition 18.** *Let* $\mathbf{X} \in \mathbb{R}^{n \times m}$*, and* $\mathbf{M} \in \mathbb{R}^{k \times m}$*, where* $k \leq m$*, and* $\mathrm{rank}(\mathbf{M}) = k$ *then if* $\exists \mathbf{A}$ *s.t.* $\mathbf{M} = \mathbf{A}\mathbf{X}$*, where* $\mathbf{A} \in k \times n$ *and* $k \leq n$*, the following holds,*

$$\mathbf{M}^\top(\mathbf{M}(\mathbf{X}^\top\mathbf{X})^+\mathbf{M}^\top)^{-1}\mathbf{M} \preceq \mathbf{X}^\top\mathbf{X}. \tag{70}$$

*Proof.* The pseudo-inverse of $(\mathbf{X}^\top\mathbf{X})^+ = \mathbf{X}^\top(\mathbf{X}\mathbf{X}^\top)^{-2}\mathbf{X}$. Consequently,

$$\begin{align}
\mathbf{M}^\top(\mathbf{M}(\mathbf{X}^\top\mathbf{X})^+\mathbf{M}^\top)^{-1}\mathbf{M} &= \mathbf{X}^\top\mathbf{A}^\top(\mathbf{A}\mathbf{X}\mathbf{X}^\top(\mathbf{X}\mathbf{X}^\top)^{-2}\mathbf{X}\mathbf{X}^\top\mathbf{A}^\top)^{-1}\mathbf{A}\mathbf{X} \tag{71} \\
&= \mathbf{X}^\top\mathbf{A}^\top(\mathbf{A}\mathbf{A}^\top)^{-1}\mathbf{A}\mathbf{X} \preceq \mathbf{X}^\top\mathbf{X} \tag{72}
\end{align}$$

where the last line follows from the properties of a projection. □