# OpenReview forum: "Experimental Design for Linear Functionals in Reproducing Kernel Hilbert Spaces"
_NeurIPS.cc/2022/Conference — NeurIPS 2022 Accept_

### Official Review · Reviewer_tM79 · 2022-07-09

**Rating:** 7
**Confidence:** 1
**Soundness:** 3 good
**Presentation:** 3 good
**Contribution:** 3 good

**Summary:**

This paper considers the problem of optimal design for estimating linear functionals in the nonparametric models restricted to the RKHS setting. Apart from optimal design points, the paper also constructs non-asymptotic confidence sets for the linear functional estimators under light-tailed noise settings.

**Questions:**

I do not have very specific comments or questions for the authors. But several papers are recommended to be cited:

1. Optimal design using SoS hierarchy: https://arxiv.org/pdf/1706.04059.pdf

2. Nonparametric function estimation: https://arxiv.org/pdf/1812.05553.pdf https://www.sciencedirect.com/science/article/abs/pii/016771528490066X

There could be more and the authors are also suggested to dig more into the literature and survey more in future versions, as I believe this could be an important service to the machine learning community.



**Limitations:**

The potential limitations are appropriately discussed.

**Strengths And Weaknesses:**

Optimal design is definitely an important topic for statistics and machine learning. The specific topic considered in this paper also tries to advance our understanding of optimal design from linear models to nonlinear models by focusing on RKHS. This is definitely a welcome contribution and might have some impact in the future.

The results are quite novel to me, in particular the part on interpolation estimators. To my best knowledge (I may be wrong here though), this seems to be the first paper considering infinite-dimensional/nonparametric models for optimal design for estimating linear functionals. There are several earlier works on optimal design for nonparametric function estimation but they did not consider linear functionals.

The theoretical results of this paper are sound and the writing is succinct and clear.

---

> ### Author Response · Authors · 2022-07-29
> **response**
>
> 1. The paper you shared is extremely interesting, however its focus is on *construction of the design* using tractable convex relaxation techniques based on SDPs. The construction is not the prime focus. Nevertheless, related but not directly. We will add it to the appendix where we discuss techniques for construction of the design.
>
> 2. We are not first to discuss non-parametric experiment design in the context of Lipschitz function (as the paper 1984 paper) nor in RKHS spaces which is the function space considered in our work (some citation in related work). We really focus on linear functional and bias associated with that as well as adaptive confidence intervals.
>
> The paper by Dette et. al. (2018) is indeed related in its focus on dependent random variables, but more restrictive than ours - they assume a Markov process, which is more restrictive than adapted sequences. Also, it uses different tools to us: continuous formulation of the problem (evaluation space are trajectories); different estimators (truncated shrinkage) and no confidence sets. However, it is perhaps worth a citation. Thank you for the reference. We were not aware of this.
>
> Did we address all your questions?

---

> > ### Comment · Reviewer_tM79 · 2022-08-07
> > **thank you for your response**
> >
> > i want to thank the authors for their response. in view of the rebuttal & my initial impression of the paper, i'm willing to increase my score to 7. but i do want to admit that i'm not sufficiently familiar with this area.

---

### Official Review · Reviewer_61h2 · 2022-07-10

**Rating:** 8
**Confidence:** 3
**Soundness:** 3 good
**Presentation:** 4 excellent
**Contribution:** 3 good

**Summary:**

This paper considers the optimal design problem for a linear model of which the slope resides in a reproducing kernel Hilbert space. Different from the vanilla linear models, regularization needs to be considered in the model studied in this paper. The observed covariance matrix of the residuals is inverted as the Fisher information matrix. A-optimum and E-optimum are considered. Confidence sets are estimated via normal and chi-squared distributions. Experimental design algorithms in the literature are discussed. My main concern is that beyond general discussions the paper may not provide sufficient new results for the high standard of NeurIPS.

**Questions:**

I find some writing problems, some rendering the relevant sections difficult for me to follow:

* the term "simple reverting law with a known gain" in Line 366.
* the term "uncertain" in Line 368.
* script "O" from Line 371 to Line 391, while this symbol was overloaded before as the big-O notation.
* some notations are not consistent. For example, C: p*m from line 41, L: p*n from line 65, X: n*n from |S|=n in line 64, but LX=C from Line 75 makes a contradiction.
* the V0 norm, in Lines 118-119 is not defined.
* A design does not maximize Tr(W), but 1/Tr(W^{-1}).
* In Lines 125 and 126, the transpose of X is left-multiplied to K, while in lines 127 and 133, it is X that is left-multiplied to K.
* I believe that the matrix V0 in Equation (5) is not defined.
* It seems that X is a matrix, but it is used as a space in Section 4.
* The concepts "support" and "allocation" need to be defined, e.g., in Proposition 5.
* In Theorems 1 and 6, matrices V-dag are overloaded to denote different objects.
* In the proof of Theorem 6, Ik and script I both represent the identity matrix.
* Instead of using an upper bound of the Chi-square probability for Theorems 1 and 2, why not just use its quantile function?
* It seems to me that Sections 7.2 and 7.3 are the focus of the paper. If this is the case, would bringing them to the beginning of the paper be better?

**Limitations:**

Limitations:
There is no obvious limitations about the methodologies studied.

Negative societal impact:
This paper is about a theoretical research, which has no direct negative societal impact.

**Strengths And Weaknesses:**

Strengths:
* comprehensive discussion of the application background
* The problem of optimal design for linear models in RHKS is important and of interest to a broad audience.
* The problems in Sections 7.2 and 7.3 appear to be new and significant applications of kernel methods.

Weaknesses:
* There might not be sufficiently new results, especially compared against the high standard of NeurIPS.

=====================================

After reading other reviews and the author(s)'s responses, I realize that this paper does have sufficient contributions to the community of optimum experimental design. The extension of experimental design problems in infinite dimensional Hilbert space seems a significant breakthrough for the community. I am happy to raise my score and am OK with the acceptance of this paper.

---

> ### Author Response · Authors · 2022-07-29
> **response**
>
> **Answers**
>
> Let us address the point in order:
> 1. If the control system is $dx(t)/dt = f(x(t)) + u(t) + \epsilon(t)$; then if we knew $f$, using a control input $u(t) = -f(x(t)) - (Kx(t))$, where the second term is known as mean reverting (mean is zero) and $K$ is known in the control literature as *gain*. If we remove the non-linearity, we can easily control the rest with the second term. It is one of the classic controllers, sometimes known as PID controller, but often when the system is very nonlinear and f is known only approximately K needs to be very big in order to force the system sufficiently quickly to equilibrium around zero. This is why we want to learn a good estimate of $f$, so that we can use a lower gain, which translates to lower sampling rate, less energy and easier physical design. Example of this simple controller in our example is explained in Section 7, line 383. The example comes from state-of-art non-linear control literature.
> 2. $A-\hat{A} = \tilde{A}$, where $\tilde{A}$ is the residual in the estimating $A$ with $\hat{A}$ and hence $\tilde{A}$ is uncertainty in estimating $A$ with $\tilde{A}$ this is why we called it uncertain. Better terminology is a $residual$.
> 3. Yes we can fix this and use a symbol $O$ without calligraphy, to denote the safe operating region
> 4. There is no contradiction, $\mathbf{X}$ is an evaluation operator, and for finite dimensional spaces its $n \times m$, not $n \times n$ as you point mention
> 5. Defined in line 24
> 6. Yes, this is a embarrassing typo in the paper we *were aware of as of the first day after the submission*, but to anybody familiar with experiment design it's clear what the right scalarization should be, and indeed in experiments we use the right scalarization. The other one is not even convex.
> 7. Indeed this is a typo. In 127 and 125, the transpose symbols should be reversed.
> 8. It is defined in line 24.
> 9. Matrix $\mathbf{X}$, is an evaluation operator $\mathcal{H} \rightarrow \mathbb{R}^n$, as explained in 70
> 10. A support of a measure over a domain is part of a domain where it is non-zero. An *allocation* is a synonym for a measure over $\mathcal{X}$ defined in 264-269, when its formal definition is needed. Informally they are defined in the first paragraph of the paper.
> 11. This is a typo, in Theorem 1, the exponent should be {-2} as in the appendix Theorem 6. It does not change the message.
> 12. They are both identity operators but in different spaces. $I_k$ on $\mathbb{R}^k$ and $\mathcal{I}$ on the studied RKHS. The notation is suggestive of this, we added clarification to the proof.
> 13. We don’t use quantile function, to get an exact dimension dependence and contrast it with the adaptive confidence sets, where this would not be possible. This is the core of our paper that you review seems to have ignored.
> 14. Focus of the paper is also Sec. 5, which is completely novel and gives rise to many interesting applications in Sec. 7 such as adaptive stopping. A simple corollary gives a novel bound on the UCB algorithm in Appendix G. We are **very strongly convinced** this is a very good contribution to which whole chapters in books are devoted (see Appendix G). The fundamental importance of adaptive confidence to the field of adaptive analysis should not be overlooked! If you know where this is solved in literature elsewhere, please provide a reference.
>
> **Contribution**
>
> Your review does not mention the actual contributions of this paper. Indeed there are some typos, and we are very grateful that you pointed them to us, but naturally any paper with 30+ pages has some typos and this is no reason to reject a paper especially since they do not change the message. We addressed the typos you mentioned and uploaded a revised draft.
>
> Our contributions are addressing the bias, motivation of optimality criteria, derivation of adaptive confidence intervals that do not scale in the dimension of the Hilbert space, and plethora of very relevant applications. We did not claim we invented fixed confidence intervals; we include those in order to show the difference to the adaptive ones. Simply in the form we state them, they do not appear in literature elsewhere. On top of that we emphasize that one has to take the bias into account when creating designs as we show in theory and in practice in Section 7. This work is a culmination of years of research into the topic and problems with linear functionals in this context have generated papers accepted in major ML conferences (see related work). We also contrast our results with the existing adaptive confidence sets for linear functionals in the literature and show that we are always tighter with the same coverage.
>
> Did we address all your questions? Please consider raising your score – this is solid work.

---

> > ### Comment · Reviewer_61h2 · 2022-08-10
> > **Thank the author(s) for the comprehensive response**
> >
> > I thank the author(s) for the comprehensive response. I have gone through the whole page of discussions. I confess that I am not an expert in experimental design, and I failed to identify the significant contributions of this paper as pointed out by the other two reviewers. I tried very hard to understand the paper which is the reason I can find many typos. Now I see that the main contribution of this paper is to extend the optimal design analysis, from Euclidean spaces to infinite dimensional spaces. For such extension, the previous easy problems, such as bias, become tough. The extension leads to fruitful applications, where I agree with Reviewer Bcdv that "example applications are both compelling and illustrative". Although I still need much study to understand some technical details (e.g. point 1 & 2 in the response to my review), I am confident now that this is a good paper for the community of optimal design. that should not be rejected.
> >
> > I did not have a concern about the writing at all. I listed some found typos but never use them as a basis for the rating (hence my "excellent" rating of writing).
> >
> > I am happy to raise my score to 8.

---

> ### Author Response · Authors · 2022-08-08
> **response**
>
> Dear reviewer,
>
> Did we address all your questions? Do you have any further questions? The time window to respond for authors is closing tomorrow.
>
> Please have a look at the Section 5 again, and the Fig. 1. The problems you identified are easily fixable typos.
>
> Please consider raising your score – this is solid work.

---

### Official Review · Reviewer_Bcdv · 2022-07-14

**Rating:** 8
**Confidence:** 3
**Soundness:** 3 good
**Presentation:** 3 good
**Contribution:** 4 excellent

**Summary:**

This paper studies the problem of estimating a linear functional of an unknown regression function lying in an RKHS, as well as optimal and adaptive design for estimating this. The paper proposes to use a ridge regression estimate together with design objectives that are variants of classical designs (e.g., $E$-, $A$-, etc. designs) that take into account the particular linear functional being estimated. The paper then derives finite-sample confidence sets for this linear functional. Finally, the paper presents methods and results for a number of practical applications with simulated data.

**Questions:**

1. Lines 24-25, “$\nu_0$… is a positive definite operator. Depending on whether $\lambda$ is known or unknown…”: I suggest explicitly clarifying that the operator $\nu_0$ is always known. I initially thought it wasn’t, and hence got confused by Eq. (3). Also, how reasonable is the assumption that $\nu_0$ is known? For example, in the case when the constraint $\theta^\intercal \nu_0 \theta \leq \lambda$ encodes Sobolev smoothness of order $s$ (i.e., $\nu_0$ is diagonal with polynomially decaying entries in the Fourier basis) this is equivalent to knowing $s$, which is usually considered unknowable in practice. EDIT: After seeing the example constraint on lines 349-351, I realized I misunderstood the role of $\theta^\intercal \nu_0 \theta \leq \lambda$. I guess I suggest adding some explanation of the role of this constraint when it is first introduced around Lines 24-25.
2. Line 28, “$C^\intercal C$ is full rank”: Why is this necessary? It seems to me that estimating $C\theta$ would only become easier if $C$ is degenerate (and the same confidence intervals would still be valid, albeit loose).
3. Line 49: What is the $f$ in $f^\intercal \Phi(x) \geq 0$? Is this some function of $\theta$? Or a typo?
4. Line 64, “Let $S \subset \cal{X}$ be a finite set of selected evaluations s.t. $|S| = n$.” This notation seems to preclude querying the same point $x \in \cal{X}$ more than once. Is this intentional? EDIT: I later found that this was discussed in Section 6. However, I still suggest clarifying this around Line 64 to prevent confusion at this point.
5. Line 67, “one classically looks at the covariance of the residuals $C\theta - Ly$: Unless I am missing something, the formula given only agrees with the usual formula for covariance if $\mathbb{E}[C\theta - Ly] = 0$, i.e., if $Ly$ is an unbiased estimate of $C\theta$. This doesn’t seem to be the case in this paper. Maybe it should say “second moment of the residuals” rather than “covariance of the residuals”?
6. The choice of kernel(/bandwidth/feature map) could have a significant impact on the derived confidence intervals; can the author provide any guidance on this choice? Also, perhaps I missed this somewhere, but I couldn’t find what feature maps $\Phi$/kernel $k$ were used for the experiments.

**Limitations:**

As far as I saw, the paper doesn't discuss statistical optimality of proposed the algorithms (e.g., in a minimax sense). I would be especially interested in how tight the confidence intervals are (relative to the tightest possible), as well as understanding the conditions under which optimizing the scalar objectives proposed in Section 3.2 efficiently optimizing their population analogues.

I don't foresee any negative social consquences of this work.

**Strengths And Weaknesses:**

This paper presents a principled and novel approach to estimating a diverse range of interesting semiparametric quantities. I think this is an interesting and understudied problem, and I appreciate the emphasis on confidence sets rather than simply deriving convergence rates. I found the main paper (but not the Appendix!) fairly well-written. I have a few questions regarding the selection of the kernel and how this impacts the derived confidence sets (see below). The example applications are both compelling and illustrative, although, if I understand correctly, they all use simulated rather than real data.

Some typos:
1. Line 44: “depends among other things” -> “depends among other things on”
2. Line 80: “change in f compared noise” -> “change in f compared to noise”
3. The Appendix seems to have a lot of grammatical and writing errors. For example, the captions of Figures 4 and 5 don’t seem grammatical.

---

> ### Author Response · Authors · 2022-07-29
> **response**
>
> **Answers**
> 1. Indeed there are multiple ways one can use the prior. Encoding the Sobolev smoothness is indeed possible as well, but for this we use the definition of the Hilbert space and the RKHS norm, this is to further finetune the regularity for certain Hilbert spaces.
> 2. This is a great question. There is a huge amount of technicalities that arise when working directly with rank deficient matrices, but indeed our point is similar to yours. Once its rank deficient just do SVD and construct the matrix with the span which reduces to the case we study. I think we had a comment about this in the earlier draft which we removed due to space constraints.
> 3. Indeed, should be $\theta$
> 4. Indeed. We expect to repeat some measurements, we’ll address this. Thank you for the feedback.
> 5. Yes *second-moment* is more appropriate
> 6. Classically, experiment design/active learning benefits from this assumption and suggests statistically good designs using this assumption. It's somewhat a chicken and egg problem without it there is not much benefit of experiment design. There are many ways to do model selection with limited initial data but all require assumptions: Mallow’s statistics, cross-validation, Bayesian evidence maximization, etc. We are assuming that we know this by some means. Usually in practice, we can specify the number of derivatives, or the specially designed kernels for our domain like in the case of strings, proteins, spatial patterns. For (partial) differential equations, the kernel is the Green’s function etc, if this is unknown we can conservatively take the sum of possible Green’s functions. Alternatively, we solved a similar problem before and we assume the same regularity.
>
> **Re Limitations:**
>
> Indeed, we are not making a claim about minimax optimality. In the context of fixed confidence sets this is perhaps clear as there is an element which achieves the second moment of residuals with equality, and there seems to be nothing that we lose on the way. This is since we are analyzing an optimal linear estimator (in a similar sense as Gauss-Markov), with optimal design. However we do not see the space to formalize this in the current manuscript.
>
> In the context of adaptively collected data, this is an active-area of research in the field of confidence sequences that is actually related to the following work arXiv:2009.03167, but even that paper does not provide a concrete answer to this question yet.
>
> Did we address all your questions?

---

> > ### Comment · Reviewer_Bcdv · 2022-08-06
> > **Thanks to the authors for their response**
> >
> > Overall, I found this paper's contributions significant and novel. I have raised my score from 7 to 8, pending further details from Reviewer 61h2 regarding why the contributions are not sufficiently new.
> >
> > I also leave the authors with a few suggestions to improve readability of the paper:
> > 1. Add some discussion (e.g., an example) motivating the constraint $\theta^\intercal \nu_0 \theta \leq \lambda$ when it is first introduced.
> > 2. Add some discussion (as in the response above) of how an appropriate kernel can be selected in practice, and clarify how this was done in the experiments.
> > 3. Proofread the appendix (e.g., the aforementioned figure captions) and add further explanatory details to the proofs in a few places (e.g., Appendix A.1) to help the reader along.

---

### Meta-Review · Area_Chair_Yupu · 2022-08-26

**Recommendation:** Accept
**Confidence:** Certain

**Metareview:**

There is a wide consensus among reviewers, after the discussion period, that this submission has strong and novel results compared to the existing literature on experimental design, which paves the way for confidence intervals that do scale in the dimension of the underlying possibly infinite dimensional Hilbert space of the problem.

**Award:**

No

---

### Decision · Program_Chairs · 2022-09-14

Accept